# Therapeutic potential of targeting microRNA-10b in established intracranial glioblastoma: first steps toward the clinic

Nadiya M Teplyuk[1], Erik J Uhlmann[1], Galina Gabriely[1], Natalia Volfovsky[2], Yang Wang[1], Jian Teng[3], Priya Karmali[4], Eric Marcusson[4], Merlene Peter[1], Athul Mohan[1], Yevgenya Kraytsberg[1], Ron Cialic[1], E Antonio Chiocca[5], Jakub Godlewski[5], Bakhos Tannous[3] & Anna M Krichevsky[1,*]

## Abstract

MicroRNA-10b (miR-10b) is a unique oncogenic miRNA that is highly expressed in all GBM subtypes, while absent in normal neuroglial cells of the brain. miR-10b inhibition strongly impairs proliferation and survival of cultured glioma cells, including glioma-initiating stem-like cells (GSC). Although several miR-10b targets have been identified previously, the common mechanism conferring the miR-10b-sustained viability of GSC is unknown. Here, we demonstrate that in heterogeneous GSC, miR-10b regulates cell cycle and alternative splicing, often through the non-canonical targeting via 5′UTRs of its target genes, including MBNL1-3, SART3, and RSRC1. We have further assessed the inhibition of miR-10b in intracranial human GSC-derived xenograft and murine GL261 allograft models in athymic and immunocompetent mice. Three delivery routes for the miR-10b antisense oligonucleotide inhibitors (ASO), direct intratumoral injections, continuous osmotic delivery, and systemic intravenous injections, have been explored. In all cases, the treatment with miR-10b ASO led to targets' derepression, and attenuated growth and progression of established intracranial GBM. No significant systemic toxicity was observed upon ASO administration by local or systemic routes. Our results indicate that miR-10b is a promising candidate for the development of targeted therapies against all GBM subtypes.

**Keywords** alternative splicing; brain tumor; microRNA; oligonucleotide therapeutics; stem cells

**Subject Categories** Cancer; Neuroscience

## Introduction

Glioblastoma (GBM), also known as grade IV astrocytoma, is the most aggressive primary brain cancer. Despite intensive treatment that includes surgery, chemotherapy, and radiotherapy, the median survival of GBM patients is around 14 months. Only a few drugs are proven to possess some therapeutic efficacy and administered for GBM patients. However, significant toxicity of those treatments and a very high rate of disease recurrence turn further care strategies to palliative.

Glioblastoma is a highly heterogeneous disease that is generally classified into four subtypes, mesenchymal, classical, proneural, and neural, based on genetic alterations, gene expression patterns, and putative cellular origin (Verhaak *et al*, 2010). Numerous mutations, expression, and epigenetic alterations occur in different GBM subtypes with frequencies ranging between 3 and 50% (Cancer Genome Atlas Research, 2008). Considering the highly heterogenic molecular landscape of GBM, identification of common regulators of tumor growth and progression across GBM is very important. High expression of microRNA-10b (miR-10b) represents a rare unifying event for gliomas, as it occurs in at least 90% of all GBM cases across all disease subtypes, as well as in low-grade gliomas (Gabriely *et al*, 2011b). Importantly, miR-10b inhibition compromises proliferation and survival of glioma cells without affecting normal neural cells cultured *in vitro*, and initial evidence also suggests the suppressive effects on tumor growth *in vivo* (Gabriely *et al*, 2011b; Dong *et al*, 2012; Lin *et al*, 2012; Guessous *et al*, 2013; Teplyuk *et al*, 2015). Therefore, considering the lack of miR-10b expression in the normal brain, targeting this molecule in GBM might represent a unique opportunity for specific and non-toxic therapy. Substantial preclinical studies *in vivo* are required to evaluate the potency and efficacy of miR-10b therapeutic targeting for GBM treatments.

1 Department of Neurology, Ann Romney Center for Neurologic Diseases, Brigham and Women's Hospital, Harvard Medical School, Boston, MA, USA
2 Simons Foundation, New York, NY, USA
3 Department of Neurology, Massachusetts General Hospital, Boston, MA, USA
4 Regulus Therapeutics Inc., San Diego, CA, USA
5 Department of Neurosurgery, Brigham and Women's Hospital, Harvard Medical School, Boston, MA, USA
*Corresponding author. Tel: +1 617 525 5195; E-mail: akrichevsky@rics.bwh.harvard.edu

miR-10b is a powerful oncogenic miRNA promoting growth and metastasis and indicative of poor prognosis in various types of cancer (Ma *et al*, 2007, 2010; Nakata *et al*, 2011; Li *et al*, 2012; Nishida *et al*, 2012; Liu *et al*, 2013b; Mussnich *et al*, 2013; Nakayama *et al*, 2013; Sun *et al*, 2013; Wang *et al*, 2013; Ouyang *et al*, 2014). Depending on the cancer type and genetic context, miR-10b acts through pleiotropic mechanisms, including control of cell proliferation, survival, migration, invasion, and epithelial-to-mesenchymal transition, and directly targeting different genes in various cells and tissues. Although several miR-10b targets have been identified in GBM and other tumors, their regulation appears highly cell- and context- specific (Ma *et al*, 2010; Gabriely *et al*, 2011a; Han *et al*, 2014; Teplyuk *et al*, 2015). For GBM, a pathology originating from genetically and epigenetically diverse backgrounds, the common mechanisms underlying miR-10b functions in survival of tumor cells are unknown.

Here, we investigated the role of miR-10b in heterogeneous GBM-initiating stem-like cells (GSC) *in vitro,* as well as in orthotopic GBM xenograft mouse models *in vivo*. GSC is a subpopulation of tumor cells with tissue stem cell properties, capable of self-renewal and providing an origin to the rest of the tumor. GSC are highly resistant to conventional chemo- and radiation therapies and associated with the disease recurrence (Bao *et al*, 2006; Liu *et al*, 2006). Therefore, development of efficient GSC targeting strategies is critically important. Patient-derived GSC growing in neurosphere cultures are highly tumorigenic; injected to athymic mice, they form aggressive and invasive tumors, more reminiscent of human GBM than glioma cell lines derived xenografts (Galli *et al*, 2004; Singh *et al*, 2004). As other glioma cells, GSC express high levels of miR-10b, and their growth is suppressed by miR-10b inhibition (Gabriely *et al*, 2011b; Guessous *et al*, 2013); however, the mechanisms underlying this growth regulation have not been identified yet. Here, we utilized GSC to investigate the mechanisms governing miR-10b-mediated survival of cancer cells *in vitro* and miR-10b potency as therapeutic target *in vivo*.

Using the whole-genome expression profiling of three distinct GSC cultures, we demonstrated that cell cycle and mRNA processing/ alternative splicing are the major cellular mechanisms commonly affected by miR-10b in GSC. Inhibition of miR-10b caused derepression of multiple mRNA targets, in most cases by non-canonical non-seed binding to their 5′UTRs. Several splicing factors were validated as direct miR-10b targets in GSC, including MBNL2 and MBNL3. Altogether, miR-10b inhibition led to a global shift in splicing patterns of GSC.

To assess the therapeutic efficacy of miR-10b inhibition for GBM, we studied the effects of anti-miR-10b treatment on GSC-derived established intracranial GBM xenograft in athymic mice and complemented this study with experiments on syngeneic GL261 glioma model in immunocompetent mice. In a series of *in vivo* experiments, we demonstrated that anti-miR-10b ASO, administered during the exponential phase of tumor growth, significantly reduced progression of established intracranial GBM. Three delivery routes for the miR-10b ASO inhibitor, including direct intratumoral injections, continuous osmotic delivery, and systemic intravenous (i.v.) injections, proved efficient in inhibiting the growth of orthotopic GBM. This study, therefore, provides a preclinical rationale for clinical evaluation of the miR-10b targeting therapies against GBM.

# Results

## GSC as a model to study miR-10b function

To characterize GSC as model to study miR-10b oncogenic function, we have determined miR-10b expression in three genetically distinct patient-derived GBM neurosphere cultures, GBM4, GBM6, and GBM8 (also referred as MGG4, MGG6, and MGG8; Wakimoto *et al*, 2009). The expression of stem cell markers, pluripotency, and tumorigenic properties of these GSC were characterized earlier (Wakimoto *et al*, 2009). The three types of GSC carry diverse genetic alterations and distinct phenotypic features and potentially represent mesenchymal and proneural subtypes (Wakimoto *et al*, 2012). We found that miR-10b was expressed in all three GSC cultures at the levels comparable to those observed in GBM cell lines, such as A172, U87, and LN229 (Fig EV1), in agreement with our observation that miR-10b is similarly expressed across various GBM subtypes in The Cancer Genome Atlas (TCGA) (Appendix Fig S1). In contrast, miR-10b expression was detected neither in primary and early passage cultures of normal human neural stem cells (NSC), nor in human astrocytes (Fig EV1).

Consistent with previous data (Guessous *et al*, 2013), inhibition of miR-10b had a strong effect on viability of GSC (Fig 1A). GSC transfected with miR-10b ASO formed neurospheres similar to control cultures, suggesting that their neurosphere-forming capacity was unaltered. However, at a later time point (day four post-transfection) the neurospheres started to exhibit markers of apoptosis followed by the massive cell death and sphere disaggregation. This process resulted in significant reduction in both number and size of the GSC neurospheres (Fig 1B). As it was previously shown for glioma cell lines (Gabriely *et al*, 2011b), miR-10b inhibition led to the cleavage of caspase 3 and caspase 7 in GSC, indicative of the induction of apoptotic cell death (Fig 1C). A significant increase in the number of apoptotic cells occurred in miR-10b depleted cultures by day five post-transfection, as also indicated by the double staining with 7-aminoactinomycin D (or propidium iodide) and Annexin V (Figs 1D and EV2). Therefore, miR-10b inhibition strongly reduced the viability of heterogeneous GSC, similarly to its effects on other subpopulations of glioma cells. At the same time, inhibition of miR-10b in differentiating GSC cultures attenuated expression of stem cell markers Nestin and OCT4 and elevated astrocytic marker GFAP (Appendix Fig S2), indicating that in differentiating conditions, miR-10b helps to maintain stem cell properties of GSC.

## MiR-10b affects cell cycle and spicing machinery in GSC through non-conventional gene targeting

To identify the common targets regulated by miR-10b in heterogeneous GSC, we transfected GBM4, GBM6, and GBM8 cultures with the miR-10b ASO or the corresponding control oligonucleotide and conducted genomewide microarrays expression profiling at 24 h post-transfection. This time frame allowed identification of direct mRNA targets and molecular pathways modulated, prior to the massive changes in gene expression associated with apoptosis. The 1,429 probe sets corresponding to transcripts and splice variants of 956 genes exhibited significantly altered expression (more than 1.2-fold, $P < 0.05$) in all three types of GSC after miR-10b inhibition.

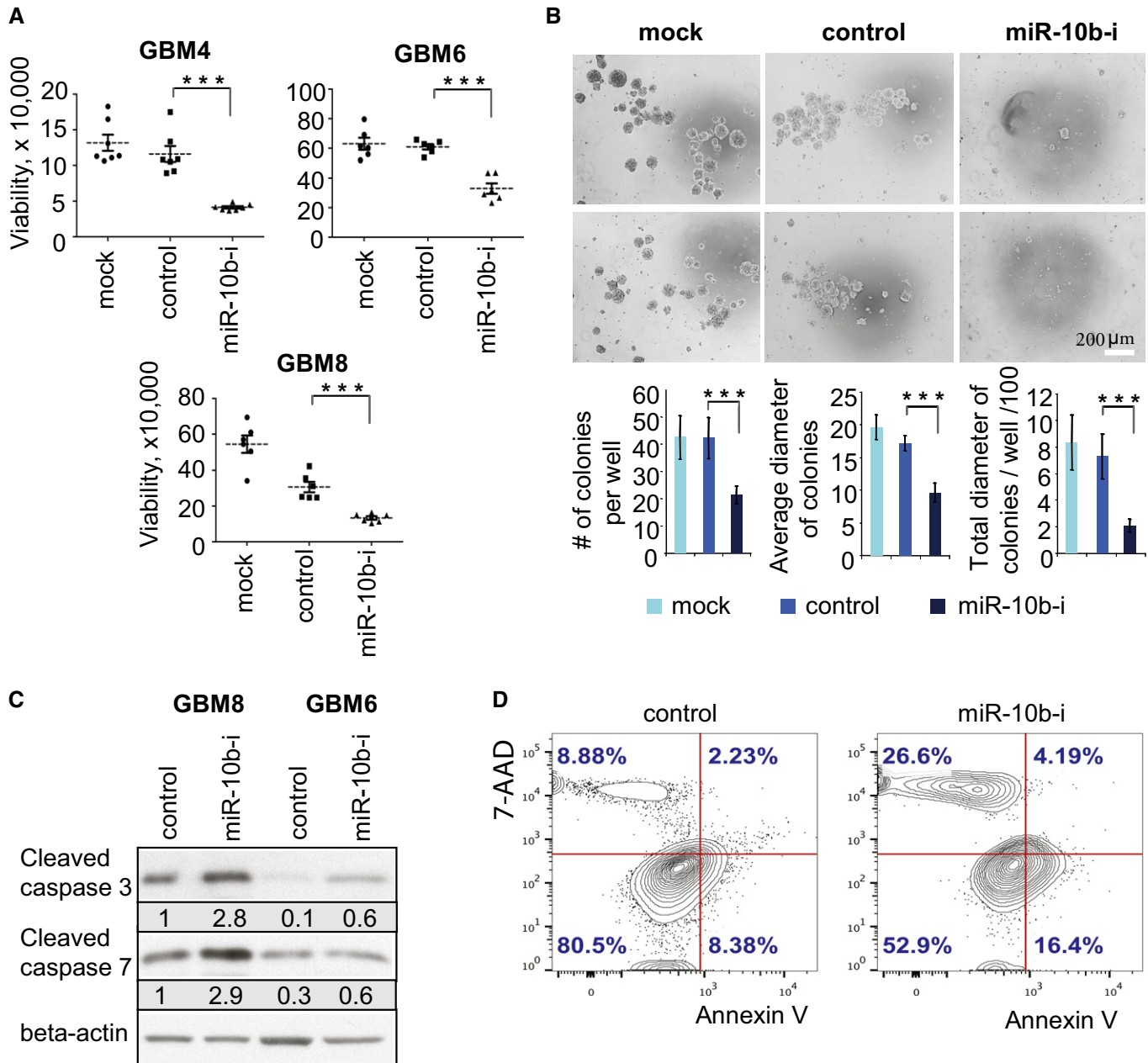

**Figure 1. MiR-10b inhibition reduces viability and enhances apoptosis of GSC.**

GSC neurospheres were dissociated to single cell suspension and transfected with either miR-10b inhibitor (labeled "miR-10b-i") or non-targeting control oligonucleotide, or treated with Lipofectamine 2000 only (Mock).

A   Cell viability was monitored at day 5 after transfection as described in Materials and Methods.

B   The number and size of GSC colonies were monitored at day 5 after transfection.

C   miR-10b inhibition induces cleavage of caspases 3 and 7 in GSC, as determined by Western blot analysis at day 5 after transfection with the inhibitor. The signals were quantified using ImageJ and normalized to beta-actin.

D   Flow cytometry analysis of Annexin V and 7-AAD staining of GSC GBM8 at day 5 after miR-10b inhibition.

Data information: (A, B) Statistical significance of the difference was determined by Student's *t*-test, with *P*-values < 0.001 indicated by three asterisks. Numbers of replicates and exact *P*-values are included in Appendix Table S4.

Source data are available online for this figure.

Five hundred and twenty-two probe sets have shown up-regulation, and 907 probe sets were down-regulated by the anti-miR-10b treatment.

Functional annotation using Ingenuity Pathway Analysis highlighted cell cycle as one of the major functions affected by miR-10b inhibition. A total of 119 genes related to bioterm "cell cycle" were

affected by miR-10b in GBM4, GBM6, and GBM8 cells (Figs 2A and EV3). This was in line with previously reported effects of miR-10b on glioma cell cycle progression (Gabriely *et al*, 2011b; Teplyuk *et al*, 2015). More detailed analysis of the expression dataset by Gene Set Enrichment Analysis (GSEA) indicated that genes related to bioterms "RNA processing" and "RNA splicing" were the most strongly enriched among the miR-10b-regulated genes in all three types of GSC (Table EV1). Expression of multiple splicing factors and components of splicosome complex were significantly affected by anti-miR-10b ASO (Fig 2B). Furthermore, alternative splice variants of numerous transcripts exhibited differential regulation based on the microarrays data (Fig EV4) and further validation for a subset of genes by isoform-specific qRT–PCR analysis (Appendix Fig S3), indicating that global splicing pattern of GSC shifted upon miR-10b inhibition. Importantly, mRNA processing and splicing-related bioterms exhibited a strong correlation with miR-10b expression in GBM TCGA dataset as well (Table EV2), substantiating the idea of miR-10b-regulated splicing in human GBM tumors *in situ*.

We next utilized the array analysis to identify common direct mRNA targets regulated by miR-10b in GSC. The classical mechanism of miRNA regulation involves the binding of 5′ miRNA "seeds" (7–8 nt motifs) to the complementary sequences within 3′UTRs of corresponding mRNA targets (Lewis *et al*, 2005), which accounts for the majority of miRNA–target interactions (Helwak *et al*, 2013). However, our computational analysis revealed no enrichment of the miR-10b seeds in the 3′UTRs of genes derepressed by miR-10b inhibition in GSC (data not shown). Emerging data suggest that miRNAs may also function via non-canonical (non-seed-mediated and/or non-3′UTR-based) mechanisms (Lytle *et al*, 2007; Chi *et al*, 2012). To explore this possibility, we scanned 5′UTRs, 3′UTRs, and complete coding sequences (CDS) of the derepressed/up-regulated genes for potential 6- to 9-nucleotide-long miR-10b complementary motifs. We found that 5′UTRs but not 3′UTRs of mRNAs up-regulated by anti-miR-10b treatment were significantly enriched in such motifs (Fig 2C). Of note, the majority of miR-10b octamers that were strongly overrepresented in the population of up-regulated transcripts (relative to the population of down-regulated or unaltered transcripts) corresponded to the miR-10b 3′ end and not to the conventional 5′ seed (Fig 2D). These results suggest that miR-10b most frequently binds to the 5′UTRs of its targets and functions via a non-conventional non-seed-mediated mechanism.

## Regulation of splicing factors by miR-10b through non-canonical binding within 5′UTR

We have further selected eight splicing factors as potential direct miR-10b targets using the following criteria: (i) Their genes were derepressed in at least two out of three GSC types by miR-10b inhibitor; (ii) expression of those genes is reduced in GBM versus the normal brain in The Cancer Genome Atlas (TCGA) or Oncomine datasets (Fig 3, Appendix Table S1), (Bredel *et al*, 2005; Liang *et al*, 2005; Lee *et al*, 2006; Sun *et al*, 2006; Murat *et al*, 2008); and (iii) corresponding mRNAs possess a putative miR-10b binding site in either 3′UTR or 5′UTR (Fig 4A). Derepression of these genes by miR-10b inhibition was validated by qRT–PCR analysis in GBM4, GBM6, GBM8, and BT74 GSC cultures, as well as additional glioma cell lines LN215 and U251 (Fig 4B). Notably, in all cases we observed a weak/moderate but statistically significant effect on these mRNAs, typical for miRNA regulation. Additional experiments on 5′UTR targets MBNL1-3, SART3, and RSRC1 demonstrated that transfections of miR-10b mimic down-regulated expression of these proteins in GSC (Fig 4C).

To validate the direct binding and regulation, 5′UTRs or 3′UTR fragments of these genes were cloned into 5′UTR LightSwitch reporter or 3′UTR PsiCheck2 reporter vectors, respectively, depending on the location of the best predicted miR-10b binding sites. We have also constructed a 5′UTR reporter of miR-10b activity by inserting a single miR-10b complementary site into the 5′UTR of luciferase gene within the LightSwitch vector. miR-10b overexpression caused a dramatic reduction in the 5′UTR reporter activity, indicating that miR-10b can significantly repress gene expression via direct binding to 5′UTR (Fig 4D). Furthermore, we found a much weaker but significant attenuation of MBNL2, MBNL3, SART3, and RSRC1 5′UTR reporter activities by miR-10b. Deletion of miR-10b binding sites completely reversed the regulation of MBNL2 and MBNL3 5′UTRs, confirming direct targeting of these splicing factors through their 5′UTRs (Fig 4E). The MBNL1 5′UTR construct did not consistently respond to changes in miR-10b levels, and miR-10b regulation of SART3 and RSRC1 constructs was incompletely rescued by mutations of the predicted miR-10b binding sites (Fig 4E). Therefore, this artificial assay has not provided firm validation of three late factors as direct miR-10b targets. Nevertheless, regulation of their mRNAs and proteins by both miR-10b inhibitors and mimics in multiple glioma cell lines

**Figure 2. miR-10b regulates cell cycle- and splicing-related genes in GSC.**

Three types of GSC (GBM4, GBM6, and GBM8) were transfected with miR-10b ASO, and gene expression was analyzed 24 h later by the Affymetrix microarrays. The heatmaps' colors intensity demonstrates altered expression of the genes (up- or down-regulated relative to the mock-treated samples) with the fold change > 1.2 and *P* < 0.05 in at least two out of the three GSC cultures.

A    The genes associated with "cell cycle" bioterm have been selected using Ingenuity Pathway Analysis. The treatment with the miR-10b inhibitor is indicated as "miR-10b-i".

B    The genes associated with "RNA splicing" bioterm have been selected using Gene Ontology (GO). The treatment with the miR-10b inhibitor is indicated as "miR-10b-i". Arrows depict the genes selected as candidate direct targets for further study.

C    miR-10b-binding motifs are enriched in 5′UTRs of the genes up-regulated by miR-10b ASO. The graph shows the probability that enrichment of the miR-10b motifs in mRNAs up-regulated vs. unchanged (*P* < 0.05) by anti-miR-10b does not occur by chance.

D    The miR-10b octamer motifs' composition of the 5′UTRs was compared between transcripts up- and down-regulated on the microarrays. The relative frequencies of various miR-10b-binding motifs are shown, indicating that mostly miR-10b 3′-end-binding motifs are enriched in the up-regulated mRNAs.

Source data are available online for this figure.

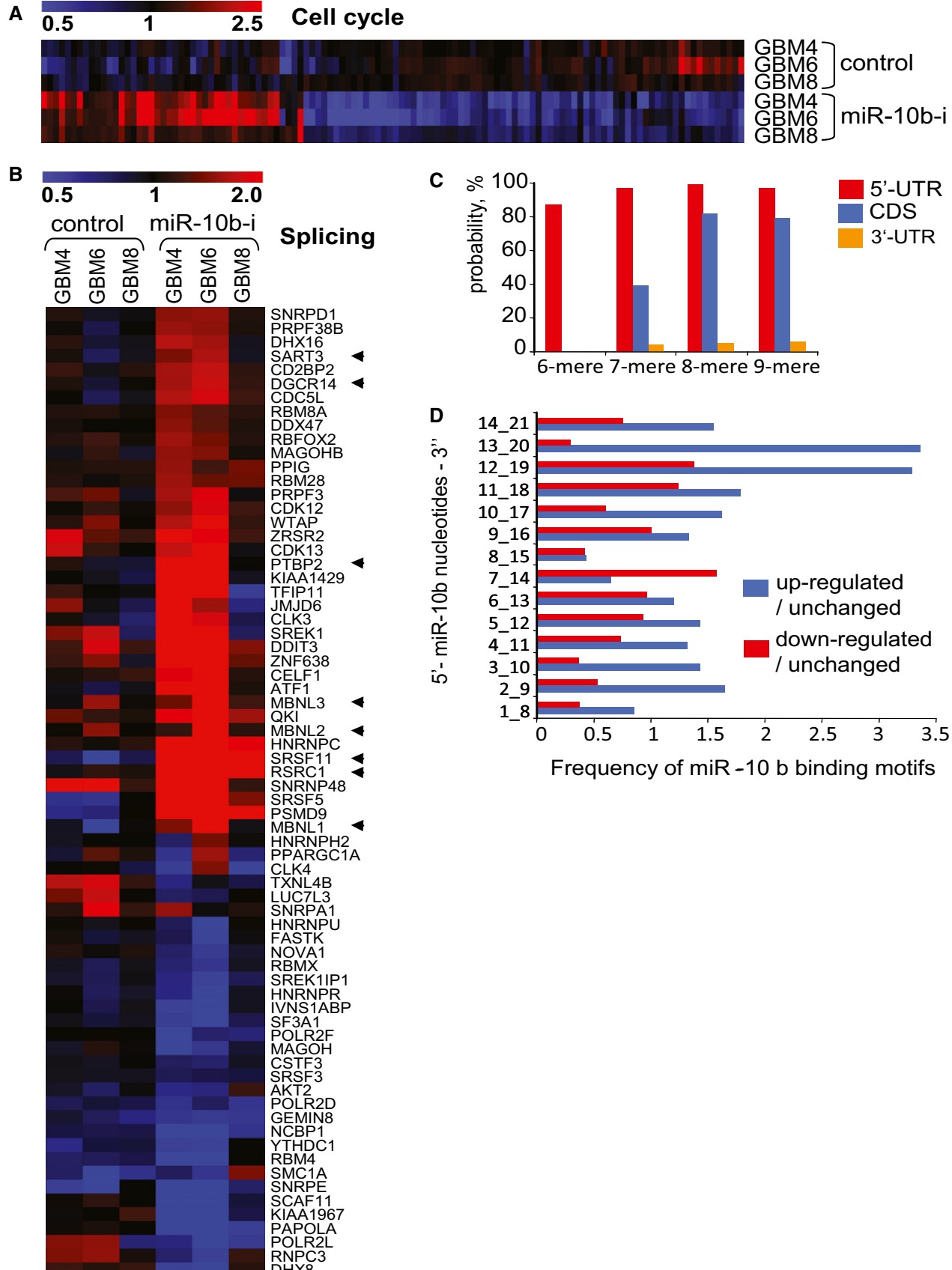

**Figure 2.**

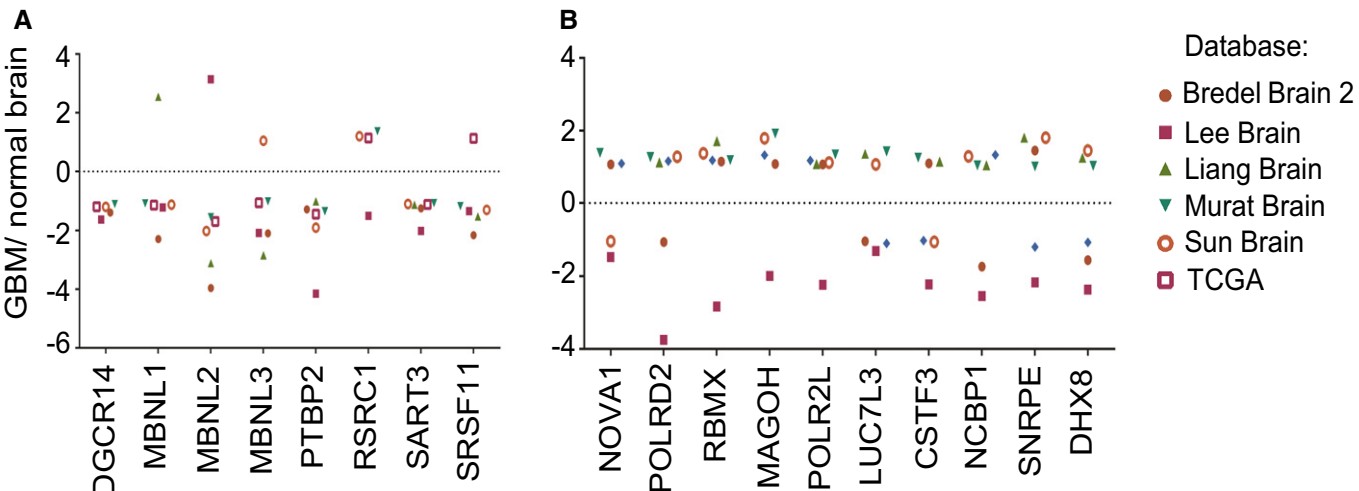

**Figure 3.   Expression analysis of splicing factor mRNAs in various GBM datasets.**

A   The genes encoding splicing factors down-regulated by miR-10b are expressed at lower levels in various GBM datasets relative to their expression in normal brain tissues.

B   In contrast, many splicing factors up-regulated by miR-10b are overexpressed in the GBM datasets.

Data information: (A, B) Six high-content GBM microarray datasets from the Oncomine resource (https://www.oncomine.org/resource/login.html), including TCGA_BrainGBM (2), Bredel Brain2 (31), Lee Brain (32), Liang Brain (33), Murat Brain (34), and Sun Brain (35), that collectively contain information for 858 GBM and 52 control samples, have been utilized for the analysis. The data is presented as log2 fold change between GBM and normal brain tissues.

Source data are available online for this figure.

(Fig 4B and C), and strong putative binding sites present in the UTRs are highly suggestive of direct regulation. miR-10b also caused a weak but significant decrease in the activity of DGCR14 and SRSF11 3′UTR luciferase reporters (Appendix Fig S4). Altogether, these results indicate that miR-10b directly binds to and fine-tunes expression of multiple splicing factors in GSC, through their 5′UTRs and 3′UTRs, via both seed- and non-seed-mediated targeting.

**Treatment with intracranially injected miR-10b inhibitor delays the progression of orthotopic GBM8 tumors**

The work from several laboratories suggests that inhibition of miR-10b reduces the growth of glioma lines and GSC, but does not affect normal neural cells, and therefore possesses strong therapeutic potential for GBM (Gabriely *et al*, 2011b; Lin *et al*, 2012; Guessous *et al*, 2013). To examine the effects of miR-10b inhibitors on intracranial GBM, we utilized orthotopic GBM8 xenograft model. This model is based on low-passage human GBM8 cells cultured as tumor neurospheres (GSC) in serum-free media, thereby retaining initial genetic and tumorigenic properties (Galli *et al*, 2004; Singh *et al*, 2004; Wakimoto *et al*, 2012). The GBM8 tumors are rapidly growing, highly diffusive, and invasive, with hemorrhagic rim and necrotic center. GBM8 cells stably expressing firefly luciferase were implanted into the striatum of the nude mice, and tumor growth was monitored by bioluminescence *in vivo* imaging. At day 20 post-implantation, when the tumors were in the exponential growth phase, miR-10b inhibitor or the corresponding control oligonucleotide of the same chemistry (2′-O-MOE with phosphodiester backbone), formulated with the *in vivo* jetPEI reagent, was delivered intratumorally by stereotaxic injections, and the injections were repeated at day 25

(Fig 5A). The efficacy of miR-10b inhibition in the tumors was confirmed by the qRT–PCR analysis (Fig 5B), and its functional outcomes further assessed by targets' derepression. The majority of miR-10b-regulated splicing factors were derepressed in tumors upon anti-miR-10b treatment (Fig 5C). Furthermore, significant inverse correlation between the expression levels of these factors and miR-10b was observed in the resected tumor tissues, confirming the efficacy and specificity of miR-10b inhibition (Fig 5D). We found that treatment with miR-10b inhibitor significantly reduced the growth rate of established and fast-growing intracranial human GBM, in comparison with the control oligonucleotide (Fig 5E and F), and prolonged mice survival (Fig 5G).

**Systemic delivery of miR-10b inhibitor reduces the growth of intracranial GBM8 tumors**

Since the blood–brain barrier is usually disrupted in GBM, which may enable the delivery of systemically administered ASO-based drugs to the intracranial tumor, we further assessed the potential of systemic anti-miR-10b treatments. In this set of experiments, we utilized 2′-O-MOE oligonucleotides with phosphorothioate backbone, since such stabilized oligonucleotides readily distribute to tissues and are taken up into cells without the need for formulations (Geary *et al*, 2015). First, the fluorescent Cy-3-labeled oligonucleotide was injected systemically (through the tail vein) to the GBM8-bearing nude mice, and its delivery to various organs tracked by fluorescence microscopy. As expected, the labeled oligonucleotide was detected in the vessels of the normal brain tissues, but not in the brain parenchyma (Fig 6A). Notably, the fluorescence signal was also detected in the intracranial tumor, suggesting a possibility of miR-10b inhibition by systemic treatments (Fig 6A).

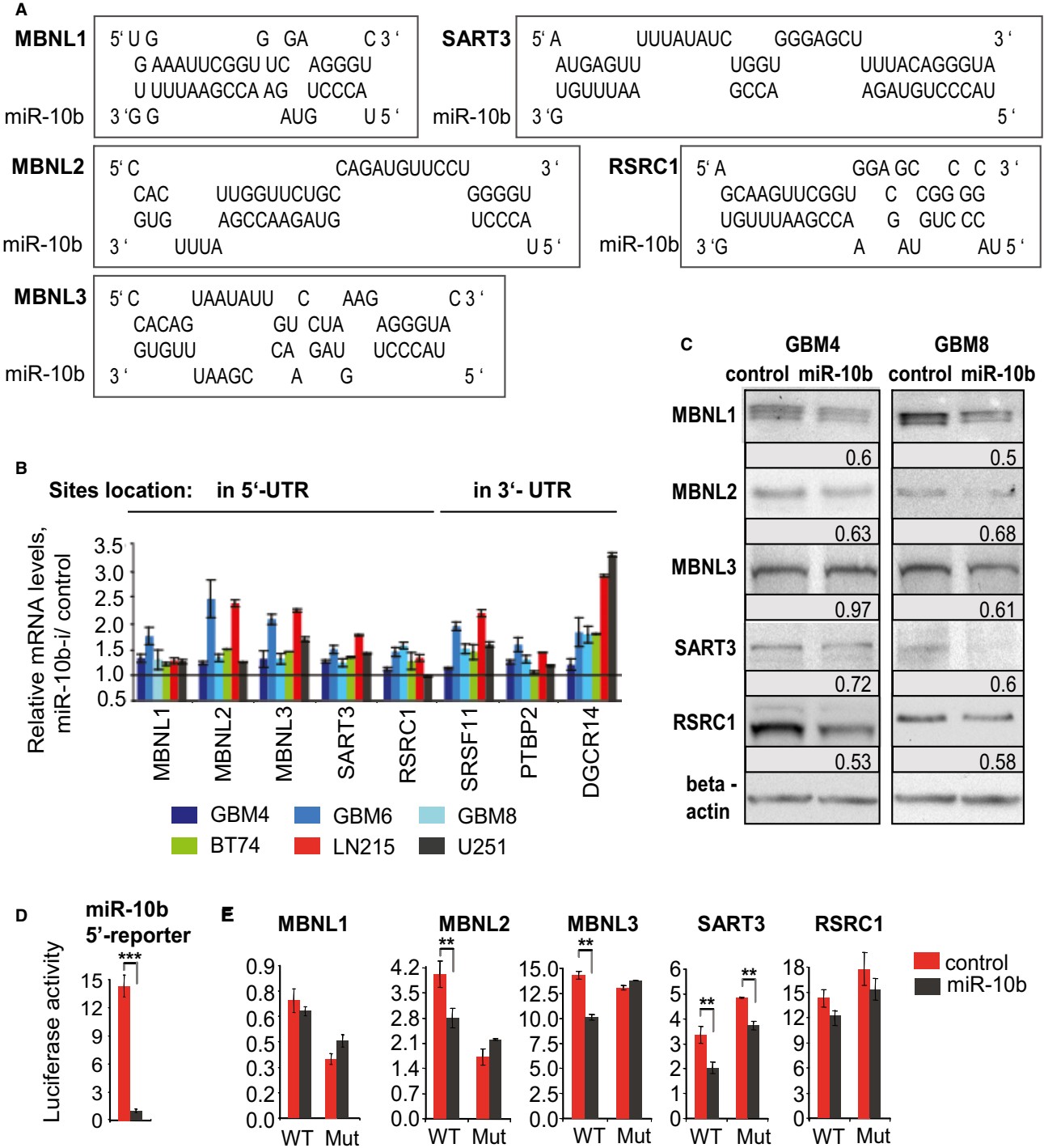

**Figure 4. miR-10b regulates splicing factors through the non-canonical binding within 5′UTRs.**

A   Putative miR-10b binding sites within 5′UTRs of candidate splicing factors mRNAs.

B   qRT–PCR analysis validates that mRNA of MBNL1-3, SART3, RSRC1, and other splicing factors are derepressed by miR-10b ASO in different GSC and GBM cell lines. mRNA expression levels were normalized to GAPDH expression.

C   Regulation of representative splicing-related proteins by miR-10b mimic in GSC, as demonstrated by Western blot analysis. The signals were quantified using ImageJ and normalized to beta-actin. The ratios between miR-10b mimic expressing and control samples are indicated.

D   miR-10b mimic regulates 5′UTR luciferase reporter containing a single miR-10b complementary site.

E   miR-10b mimic regulates 5′UTR luciferase reporters of some splicing factors genes bearing wild-type (WT) but not mutated (Mut) miR-10b binding sites.

Data information: (B, D, and E) Statistical significance of the differences was determined by Student's $t$-test, $**P < 0.01$ and $***P < 0.001$. Numbers of replicates and exact $P$-values are included in Appendix Table S4.

Source data are available online for this figure.

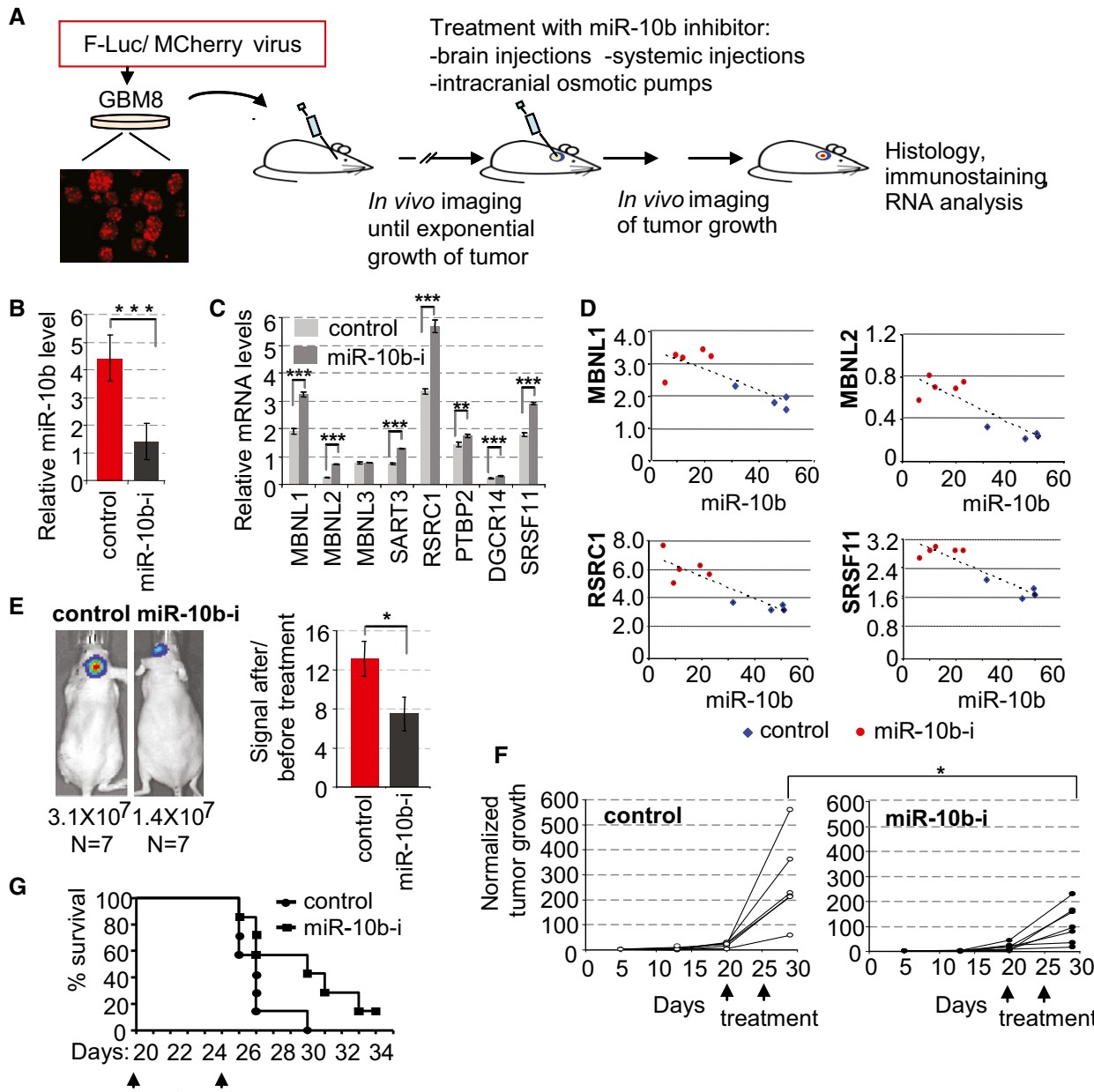

**Figure 5.  Intratumoral injections of miR-10b inhibitor reduce the growth of established intracranial GBM8 xenografts.**

A   A schematic overview of *in vivo* experiments on orthotopic GBM8. The tumor growth was monitored by luciferase imaging (WBI) and expressed in photon flux per second. Mice assigned to the treatment and control groups were treated with miR-10b inhibitors or corresponding control oligonucleotides in different formulations.

B   2′-O-MOE-PO miR-10b inhibitor (miR-10b-i) or non-targeting control (1 μg of each) formulated with *in vivo* jetPEI were injected intratumorally at days 20 and 25 after cells implantation. The efficacy of miR-10b inhibition was assessed by qRT–PCR analysis of the resected tumors, with miR-10b expression levels normalized to miR-125b.

C   qRT–PCR analysis demonstrates that miR-10b inhibition in orthotopic GBM8 leads do derepression of its mRNA targets. mRNA expression levels were normalized to GAPDH.

D   Inverse correlation between miR-10b levels and expression of its mRNA targets in resected GBM8 tumors.

E   Inhibition of miR-10b markedly reduces tumor burden. The left panels illustrate tumor imaging in representative animals at day 29. The bars represent average signal ratios for each group at day 29 (after treatment) to day 20 (at the beginning of treatment). *N* = 7 animals per group at treatment initiation.

F   Growth curves of individual tumors, based on the ratios of bioluminescence signals to the baseline signals at day 5.

G   Each mouse was sacrificed when the tumor-generated signal reached $1.5 \times 10^{7}$ photons/s, and Kaplan–Meier survival plots were built retrospectively.

Data information: (B, C, E, and F) Statistical significance of the differences was determined by Student's *t*-test, with *$P < 0.05$, **$P < 0.01$, and ***$P < 0.001$. Numbers of replicates and exact *P*-values are included in Appendix Table S4.

Source data are available online for this figure.

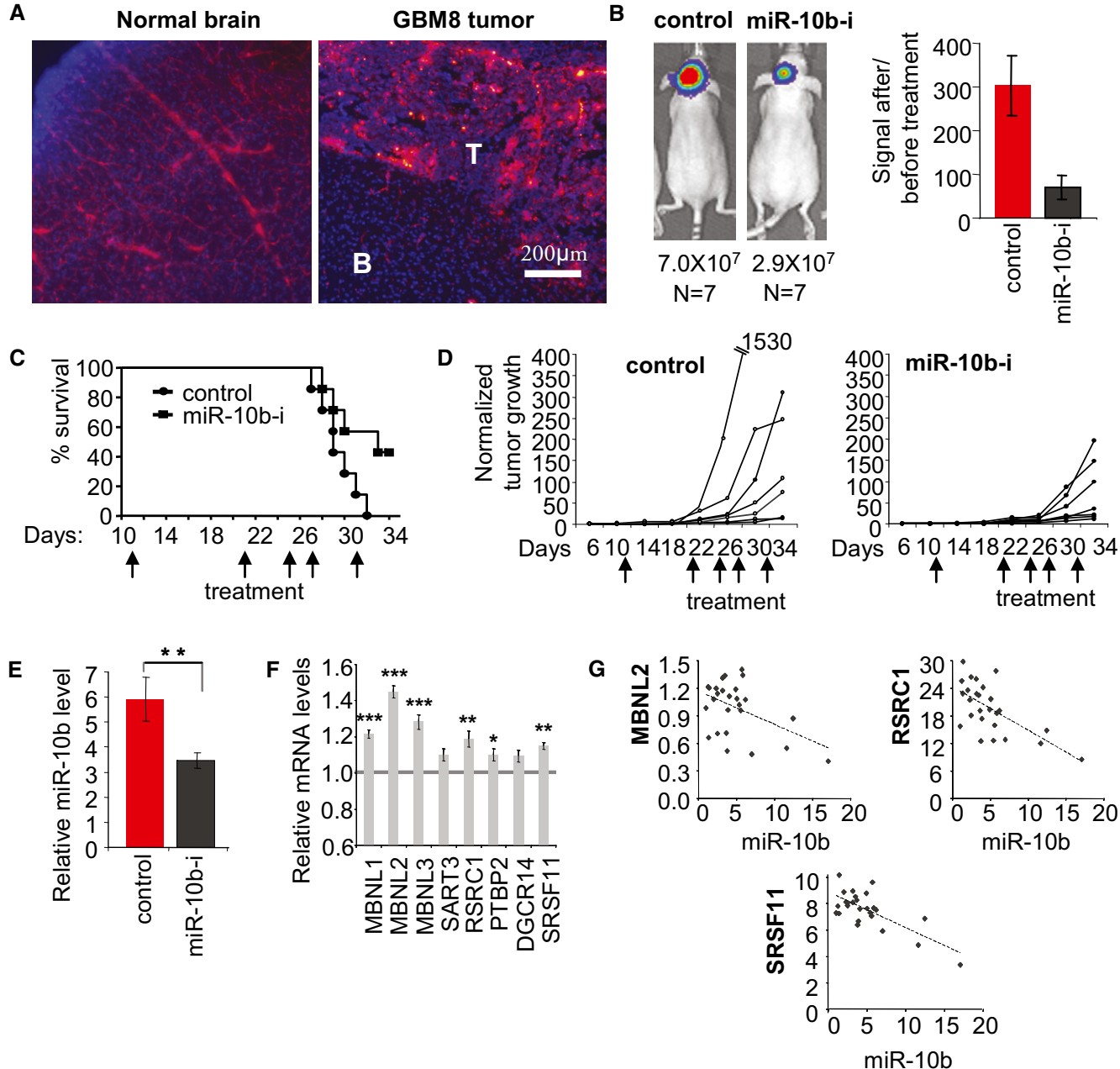

**Figure 6. Systemic treatment with miR-10b inhibitor reduces the growth of established intracranial GBM8 tumors.**

A    Intravenously injected Cy5-labeled 2′-O-MOE-PS oligonucleotide is distributed to intracranial GBM8 tumor. In the normal brain, the signal is observed in blood vessels but not within brain parenchyma. "T"—tumor, "B"—brain tissue. Each image is representative of three mice analyzed.

B–D    Systemic inhibition of miR-10b markedly reduces tumor burden. Uncomplexed 2′-O-MOE-PS miR-10b inhibitor (miR-10b-i) or non-targeting control of the same chemistry was injected at 80 mg/kg through the tail vein at the days indicated by arrows. (B) The left panels illustrate tumor images of representative animals at day 34, and average signals (photons/sec) are indicated below the images. The bars represent average signal ratios for each group at day 34, relative to day 6. (C) Each mouse was sacrificed when the tumor-generated signal reached $5 \times 10^7$ photons/sec, and Kaplan–Meier survival plots were built retrospectively. (D) Growth curves of individual tumors based on the ratios of bioluminescence signals to the baseline signals at day 6.

E    The efficacy of miR-10b inhibition in intracranial tumors was assessed by qRT–PCR analysis of the resected tumor tissues, with miR-10b expression levels normalized to miR-125b.

F    qRT–PCR analysis demonstrates that miR-10b targets were derepressed in orthotopic GBM8 upon systemic administration of the miR-10b inhibitor. Seven tumors per condition and two specimens per tumor have been analyzed. mRNA expression levels were normalized to GAPDH.

G    Inverse correlation between miR-10b levels and expression of its mRNA targets in resected GBM8 tumors.

Data information: (B, E, and F) Statistical significance of the differences was determined by Student's *t*-test, with *$P < 0.05$, **$P < 0.01$, and ***$P < 0.001$. Numbers of replicates and exact *P*-values are included in Appendix Table S4.

Source data are available online for this figure.

GBM8-bearing mice were treated with anti-miR-10b or control ASO by five systemic injections (80 mg/kg) over the 20-day period. As it was observed with intratumoral injections, systemic administration of miR-10b ASO strongly reduced the rate of tumor growth, and increased survival relative to the control group (Fig 6B–D). Analysis of RNA extracted from tumor tissues revealed almost twofold reduction in miR-10b levels in the treatment group (Fig 6E), as well as slight derepression of miR-10b target genes (Fig 6F). Furthermore, inverse correlation between the levels of miR-10b and mRNAs for target splicing factors was observed among the tumor tissues treated with miR-10b ASO and control oligonucleotide (Fig 6G). Importantly, although high amounts of the 2′-O-MOE-PS oligonucleotide were detected in various extracranial organs such as liver and kidney (Fig 7A), no apparent systemic effects, weight loss, or abnormal tissue morphology was observed during the 2 weeks of systemic anti-miR-10b treatment (Fig 7B–D). Therefore, systemically administered miR-10b inhibitor can be delivered to GBM, possibly through the disrupted blood–brain barrier, and delay the progression of actively growing tumors.

### Continuous delivery of miR-10b inhibitor by osmotic pump reduces proliferation and increases apoptosis in orthotopic GBM8 tumors

We have also examined the effect of continuous treatment with miR-10b ASO on intracranial GBM. For this purpose, we utilized convection-enhanced delivery (CED) through the intracranial osmotic pumps. For continuous delivery, the drug formulation should be stable over the period of treatment, and at the same time compatible with osmotic delivery. In these experiments, we utilized miR-10b ASO of PO chemistry, formulated with specifically designed cationic lipid nanoparticles. Osmotic pumps preloaded with either anti-miR-10b-containing nanoparticles or non-targeting control oligonucleotide-containing nanoparticles were implanted to the mice with catheters inserted intratumorally. The nanoparticles were administered to the intracranial tumors starting from day 26 after cell implantation (2 μg of ASO daily), when the tumors reached a relative luciferase signal of about ~$10^6$ photons/s and exhibited exponential growth. Continuous delivery of miR-10b ASO for 2 weeks significantly reduced progression of intracranial GBM in comparison with control treatment (Fig 8A). Histological examination of the tumor tissues revealed the decrease in cell proliferation in the treatment group (as it is evident by decreased staining for proliferation markers PCNA and KI67) (Fig 8B, C and F) and increase in apoptosis (cleaved caspase 3 staining, Fig 8D and F). No significant difference in tumor cells migration and invasion was observed between the treatment and control groups (Fig 8E and F).

Therefore, various formulations of miR-10b ASO delivered to orthotopic GBM via three routes (direct intratumoral injections, systemic injections, and osmotic pumps) reduced the progression of established and highly aggressive GBM8 tumors.

### Continuous treatment with miR-10b inhibitor reduces the growth of intracranial GL261 glioma in immunocompetent mice

Orthotopic GBM xenograft models derived from human glioma cells such as GBM8 utilize immunocompromised (athymic) mice as

recipients. To determine whether anti-miR-10b treatment can also inhibit GBM progression in immunocompetent animals, we have utilized highly aggressive mouse GL261 glioma allografts implanted into syngeneic Black 6 Albino mice. Cultured GL261 cells express high level of miR-10b (Fig EV1), and inhibition of miR-10b significantly reduces the viability of GL261 cells *in vitro*, similarly to the effect observed in human GSC (Fig 9A and B). Established intracranial tumors derived from the luciferase-expressing GL261 cells were treated by continuously delivered miR-10b inhibitor or control oligonucleotide, with osmotic pumps implanted as described above. Both miR-10b inhibitor and control oligonucleotide were efficiently delivered to the tumor tissues, as it is evident by tissue immunostaining with specific antibodies recognizing oligonucleotides with phosphorothioate backbone (Fig 9C). The miR-10b inhibitor significantly reduced the growth of GL261 tumors (Fig 9D). The efficacy of miR-10b inhibition and derepression of its previously validated direct target p21 (Gabriely *et al*, 2011b) were confirmed by qRT–PCR analysis (Fig 9E and F).

We further tested the effects of systemic delivery of miR-10b inhibitor to orthotopic GL261 tumors. The animals were injected subcutaneously (s.c.) with a daily dose of 100 mg/kg 2′-O-MOE-PS miR-10b ASO, which led to apparent delivery of miR-10b inhibitor to growing GL261 tumors. However, the efficacy of systemic delivery to GL261 tumors was significantly lower than to GBM8 tumors, with only around 10% of GL261 cells positive for the oligonucleotide (Fig EV5A), in comparison with 70–80% of positive GBM8 cells (Fig 6A). This is most likely due to the more compact and less invasive structure of GL261 tumors. Tumor growth rate and mouse survival were slightly but insignificantly affected by miR-10b ASO relative to non-specific control (Fig EV5C), which was expected considering the low uptake of the systemic ASO by orthotopic GL261. Importantly, we have not observed any systemic toxicity of high-dose miR-10b ASO in immunocompetent mice (Fig EV5B).

## Discussion

miR-10b, the miRNA most highly up-regulated in GBM, is a potent oncogenic molecule controlling cell cycle and survival of GBM cells (Gabriely *et al*, 2011b). All glioma cell types investigated thus far exhibit the hallmark high levels of miR-10b, and this miRNA was proposed as a biomarker for diagnostics and monitoring of GBM (Teplyuk *et al*, 2012). Inhibition of miR-10b leads to the apoptosis of glioma cells of all GBM subtypes. GBM is a highly heterogeneous disease, driven by multiple molecular alterations in several signaling pathways; therefore, many drugs developed for GBM might be efficient only for a subset of patients. However, due to its unique expression pattern and functional properties, miR-10b represents a promising therapeutic target common for most if not all GBM cases. Of note, although a few miR-10b targets in glioma have been previously reported, their regulation appeared highly cell- and context-specific (Gabriely *et al*, 2011a; Lin *et al*, 2012; Teplyuk *et al*, 2015). Therefore, despite the significant association of miR-10b with glioma viability, a common mechanism of miR-10b function remained unclear. The goal of this work was therefore twofold: to investigate the common molecular mechanism underlying miR-10b function in glioma, and explore the therapeutic potential of its targeting in orthotopic GBM models.

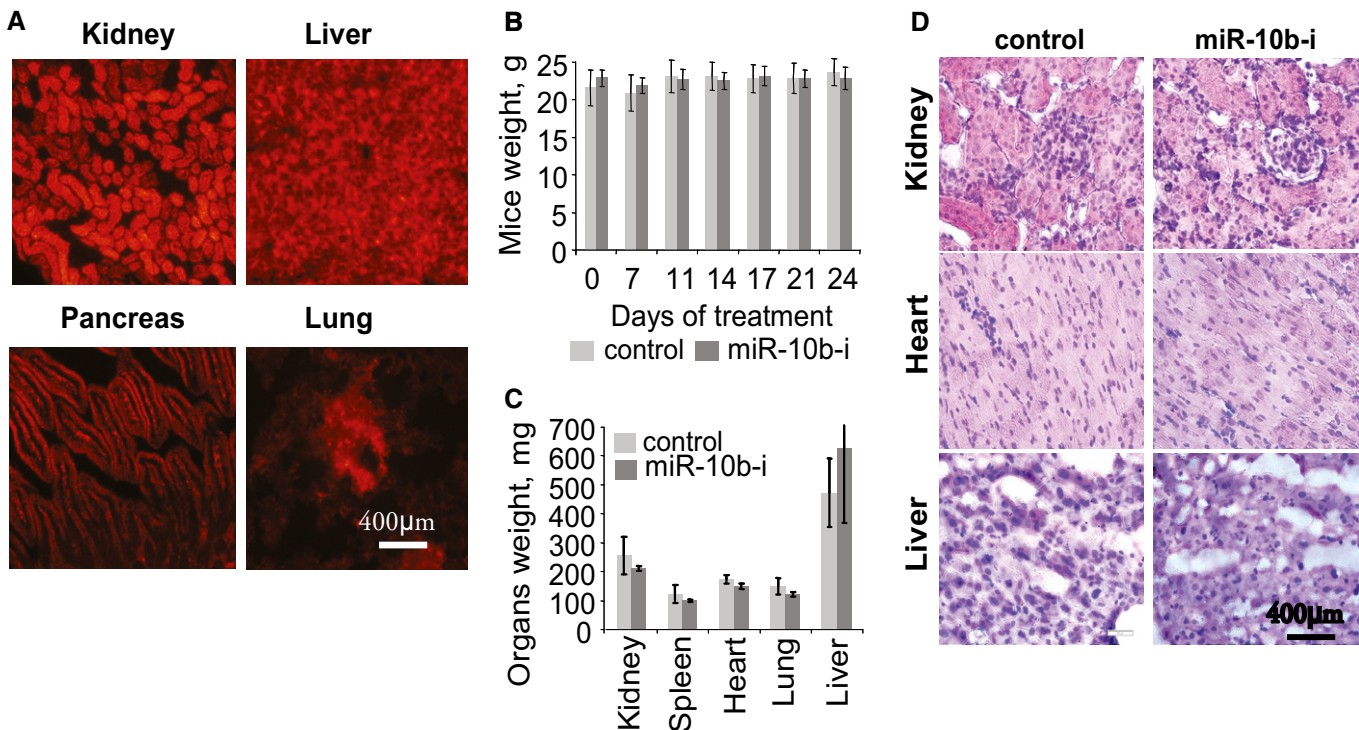

**Figure 7.  Toxicity assessment of the systemic treatment with miR-10b inhibitor.**

A      Uptake of the uncomplexed Cy5-labeled 2′-O-MOE-PS oligonucleotide (80 mg/kg injected via the tail vein) by normal extracranial tissues was examined by fluorescence microscopy 24 h after injections.

B–D   Systemic treatment of intracranial GBM8 tumors with uncomplexed 2′-O-MOE-PS miR-10b inhibitor (miR-10b–i) or non-targeting control oligonucleotide (at 80 mg/kg) was not associated with toxic effects. (B) No significant difference in average mice weight was observed between the anti-miR-10b and control treatment groups. (C) No significant difference in average organs' weight was observed between the anti-miR-10b and control treatment groups. (D) No significant difference in tissue histology using hematoxylin and eosin staining was observed between the anti-miR-10b and control treatment groups. The error bars (in B and C) represent Standard Deviation within each group of mice, $N = 7$ mice per group.

Source data are available online for this figure.

Glioma stem cells give rise to the bulk of the tumor and represent the most treatment-resistant population of cancer cells associated with tumor recurrence (Bao *et al*, 2006; Liu *et al*, 2006; Rich & Bao, 2007). Consistent with other reports, we found that miR-10b is highly expressed in these cells, while silent in normal neuroprogenitors (Lang *et al*, 2012; Guessous *et al*, 2013), suggesting that miR-10b induction might be a key early event in gliomagenesis. In this study, we employed three heterogeneous patient-derived cultures of GSC that exhibit a range of genetic and molecular aberrations and represent different GBM subtypes (Wakimoto *et al*, 2012). Specifically, GBM4 GSC are characterized by MYC amplification, GBM6—by EGFR and MDM4 amplification and CDKN2A&B deletion, and GBM8—by PDGFRA and MDM2 amplification and CDKN2A&B deletion. All GSC types appeared highly sensitive to the inhibition of miR-10b, which substantially reduces their viability, and tumor-initiating properties. Inhibition of miR-10b led to the GSC death that was observed by several markers starting from day 3 or 4. In differentiation conditions, miR-10b inhibition attenuated expression of stemness markers and promotes GSC differentiation. Supported by prior observations (Gabriely *et al*, 2011b; Guessous *et al*, 2013), this data indicate a firm requirement for miR-10b in GSC self-renewal, maintenance, and survival, regardless of the tumor subtype.

To investigate the common direct targets of miR-10b, and early events that lead to GSC death upon miR-10b inhibition, we performed whole-genome expression profiling 24 h after miR-10b inhibition. At this time point, no morphological changes or pro-apoptotic marks have yet been observed in miR-10b-inhibited cultures. As expected from the previous studies (Gabriely *et al*, 2011b; Teplyuk *et al*, 2015), our analysis identified numerous cell cycle-related genes regulated by miR-10b. Of note, however, is that validated miR-10b targets and cell cycle inhibitors CDKN2A/p21 and CDKN1A/p16 were not uniformly expressed and not always affected in the GSC analyzed. Expression levels of several other miR-10b targets previously validated in a specific U87-derived mesenchymal cell line (Lin *et al*, 2012) were also unaffected in GSC. We reasoned, therefore, that the apoptotic cell death of GSC caused by anti-miR-10b was mediated through a different set of targets. Here, we demonstrate that alternative splicing is another major function commonly affected by miR-10b in GSC. miR-10b inhibition caused a global shift in splicing pattern of numerous genes. Further-more, expression of many RNA-binding proteins involved in splicing appeared regulated by miR-10b; among them, we validated several regulators of alternative splicing as direct miR-10b targets. These new targets included the MBNL family (MBNL2 and MBNL3), SART3, RSRC1, SRSF11, PTBP2, and DGCR14. Importantly, based

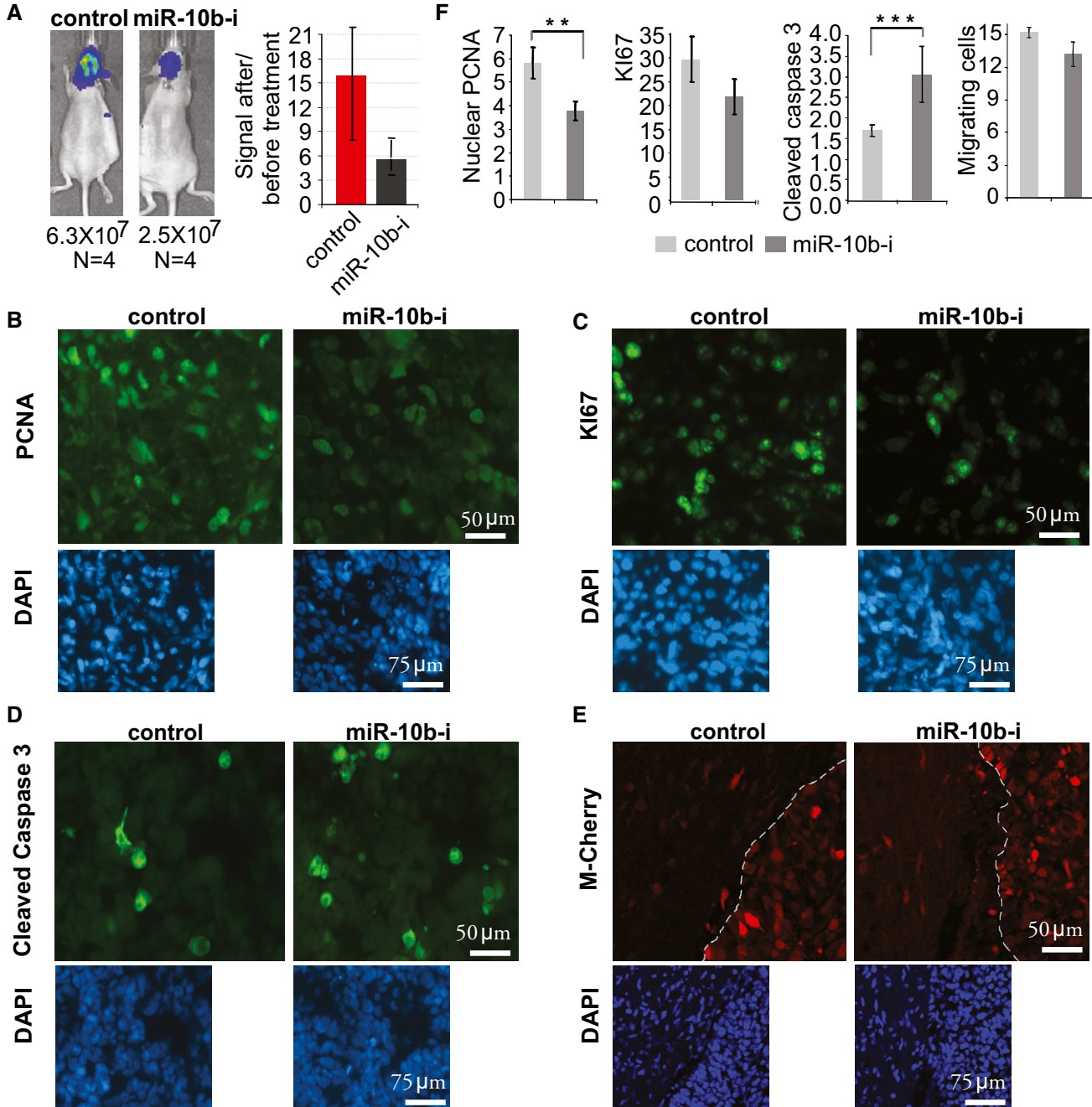

**Figure 8.  Continuous osmotic delivery of miR-10b inhibitor reduces the growth of established orthotopic GBM8 tumor xenografts.**

A   Continuous osmotic delivery of miR-10b inhibitor markedly reduces tumor burden. The osmotic pumps, loaded with lipid nanoparticles formulated with 2′-O-MOE-PO miR-10b inhibitor or non-targeting control, infused 2 μg of the oligonucleotides per day intratumorally, over 13 days. Tumors growth was monitored by the WBI, and the left panels illustrate tumor imaging of representative animals at the end of treatment. The bars represent average signal ratios for each group at day 13 (end of treatment), relative to day 2 (treatment initiation).
B   Representative immunostaining of tumors for PCNA proliferation marker.
C   Representative immunostaining of the tumors for KI67 proliferation marker.
D   Representative tumor immunostaining for cleaved caspase 3 as a marker of apoptosis.
E   Tumor cell invasion was examined by fluorescence microscopy for mCherry-positive cells migrating through the tumor border.
F   Quantitative immunostaining analysis indicates that proliferation and apoptosis markers are affected by anti-miR-10b treatment. No significant effect on invasion of intracranial GBM8 was observed.

Data information: (B-E) For each staining, the immunopositive area was quantified and normalized to DAPI-stained area using ImageJ software. The 40–50 microscopic fields were quantified within four sections per each tumor, and average values of four tumors per group are presented. (F) Statistical significance of the differences was determined by Student's *t*-test, with **$P < 0.01$ and ***$P < 0.001$. Numbers of replicates and exact *P*-values are included in Appendix Table S4.
Source data are available online for this figure.

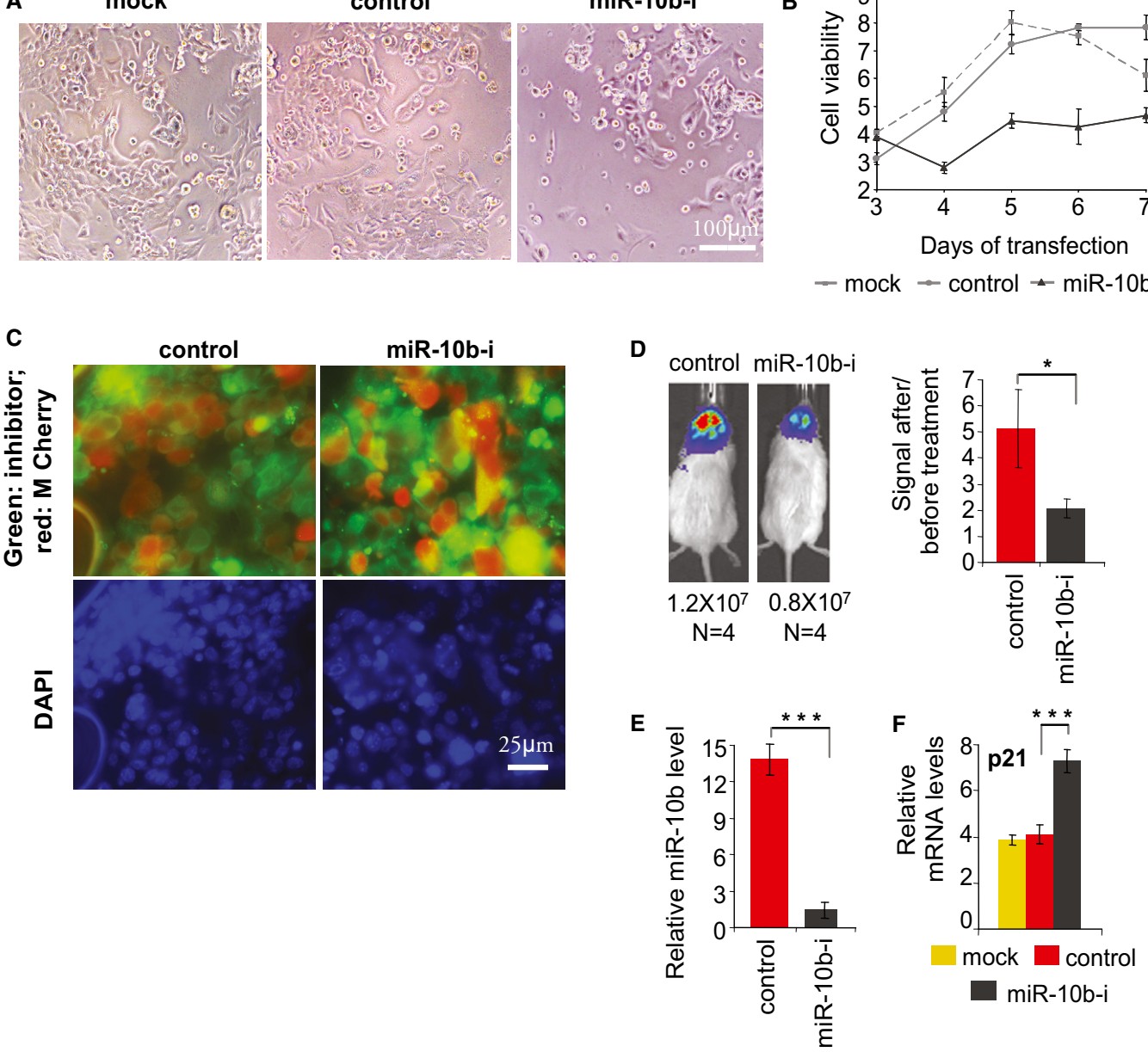

**Figure 9.  miR-10b inhibition reduces the growth of mouse GL261 glioma cells *in vitro* and GL261-derived intracranial tumors in immunocompetent mouse model.**

A, B  miR-10b inhibition decreases GL261 cell viability. Cell viability was measured at days 3–7 after transfection with miR-10b inhibitor, non-targeting control, or Lipofectamine 2000 alone (mock). (A) Phase-contrast photographs of GL261 cultures at day 6 post-transfection. (B) Growth curves of cultured GL261 cells, based on the viability assay.

C  *In vivo* jetPEI-formulated 2′-O-MOE-PS/PO miR-10b inhibitor or non-targeting control was infused to orthotopic GL261 tumors by osmotic pumps, starting at day 6 after cell implantation. The uptake of ASOs was confirmed by IHC for PS-containing oligonucleotides (green). GL261 tumor cells expressing M-Cherry are red.

D  Osmotic delivery of miR-10b inhibitor markedly reduces GL261 tumor growth in immunocompetent Black 6 Albino mice. Mice photographs show tumor imaging in representative animals at day 3 after pump implantation, and average signals in photons per second are indicated. Tumor growth rates were calculated as ratios of the signals at day 3 of the treatment to day 1 prior to initiation of the treatment.

E  The efficacy of miR-10b inhibition in intracranial tumors was assessed by qRT–PCR analysis of the resected tumor tissues, with miR-10b expression levels normalized to miR-125b.

F  qRT–PCR analysis demonstrates that miR-10b target p21 was derepressed in GL261 tumors upon miR-10b inhibition. mRNA expression levels were normalized to GAPDH.

Data information: (B, D-F) Statistical significance of the differences was determined by Student's *t*-test, with *$P < 0.05$ and ***$P < 0.001$. Numbers of replicates and exact *P*-values are included in Appendix Table S4.

Source data are available online for this figure.

on the large-scale public datasets including TCGA, all those proteins are down-regulated in GBM relative to the non-neoplastic brain tissues (Fig 3), suggesting that they are repressed by miR-10b in human tumors, and their reduced levels contribute to the dysregulated gene expression in GBM. Moreover, expression of three of them, MBNL1, MBNL2, and DGCR14, showed significant inverse correlation with miR-10b in the GBM TCGA.

Switching between splicing isoforms is one of the key cell identity-defining mechanisms, regulating stem cell and pluripotency versus differentiation state (Gabut *et al*, 2011), carcinogenesis (Venables *et al*, 2009), and organism development, reviewed by Jangi and Sharp (2014) and Kalsotra and Cooper (2011). The MBNL family of splicing factors regulated by miR-10b in GSC is a potent master switch inhibiting the stem cell phenotype and promoting differentiation. They regulate approximately half of embryonic stem cell (ES cells)-specific alternative splicing events (Han *et al*, 2013). Knockdown of MBNL family proteins in differentiated cells causes a shifts to stem cell-specific splicing pattern, enhances expression of key pluripotency genes, and significantly promotes somatic cells reprogramming to induced pluripotent stem cells (Han *et al*, 2013). Therefore, even the slight repression of these factors by miR-10b observed in GSC might play a tumor-promoting role, and enable these tumor-initiating cells to retain their stemness. Importantly, it was recently found that MBNL family regulates the splicing of neurofibromin 1, a factor playing the central role in initiation and progression of mesenchymal GBM (Fleming *et al*, 2012). Another miR-10b-regulated splicing factor, SART3, is known as a tumor-rejection antigen. It activates cytotoxic T lymphocytes reacting against a variety of tumors, including gliomas (Murayama *et al*, 2000). Therefore, the down-regulation of SART3 by miR-10b might contribute to the immunosuppression observed in GBM. SRSF11, a multifunctional nuclear protein also regulated by miR-10b, among its other activities triggers alternative splicing of TERT, telomerase reverse transcriptase, gene critical for cancer stem cells self-renewal (Listerman *et al*, 2013). The expression of polypyrimidine-tract-binding protein PTBP2 (also called neural PTB, or nPBT) increases during neuronal differentiation, reprogramming the splicing pattern to neuronal-specific (Boutz *et al*, 2007; Makeyev *et al*, 2007). Inhibition of PTBP2 by miR-10b might, therefore, also contribute to GSC maintenance in their undifferentiated state. Although the effects of miR-10b on each of those targets are relatively weak and consistent with the fine-tuning model of miRNA regulation (Bartel, 2009), cumulatively their orchestrated regulation might redefine the cell fate of glioma cells.

The expression profiling of miR-10b-affected genes also provided important insights into unconventional mechanism of miR-10b targeting. In accordance with earlier observations that miR-10b rarely regulates its predicted targets (Gabriely *et al*, 2011b; Guessous *et al*, 2013), and our unpublished data), we detected no enrichment of conventionally predicted miR-10b-binding motifs (based on perfect pairing of nucleotides 2–7, known as the miRNA seed) among 3′UTRs of the genes regulated by miR-10b in GSC. However, various miR-10b-binding motifs were significantly enriched in the 5′UTRs of the regulated genes. This enrichment was detected only when all putative miR-10b-binding motifs, including those pairing with miR-10b 5′ seeds and its 3′ region, were considered. Therefore, miR-10b interactions with its targets appear less seed-driven than those exhibited by most other miRNAs, with the

compensatory binding via miR-10b 3′ region playing a more significant role. Furthermore, miR-10b has a clear preference to bind and regulate its targets through their 5′ UTRs.

Although miRNA interactions with and regulation of mRNA expression through 5′UTR regions have been observed (Lytle *et al*, 2007; Orom *et al*, 2008; Helwak *et al*, 2013), overall such regulation is relatively rare. Based on the recently developed crosslinking, ligation, and sequencing of hybrids technique (CLASH) (Helwak *et al*, 2013), they account for 3.8% of all miRNA interactions. In addition to the miRNA-mediated posttranscriptional repression (Lee *et al*, 2009; Grey *et al*, 2010; Moretti *et al*, 2010; Zhou & Rigoutsos, 2014), direct target up-regulation through 5′UTR interaction has been reported (Tsai *et al*, 2009; Liu *et al*, 2013a; Panda *et al*, 2014). Along with translational, a transcriptional regulation by miRNA through 5′UTR has been also observed (Liu *et al*, 2013a). Of note, miR-10a, the close homolog of miR-10b distinct only in a single nucleotide, is capable of enhancing translation of ribosomal protein mRNAs by binding to theirs 5′UTRs (Orom *et al*, 2008). Nevertheless, we have not detected any effects of miR-10b on ribosomal proteins in glioma cells (data not shown). Several studies suggest that a single miRNA binding site within a 5′UTR may be insufficient for miRNA regulation (Lytle *et al*, 2007; Moretti *et al*, 2010). Our work demonstrates that a single site fully complementary to miR-10b within 5′UTR confers a powerful repression of a reporter gene. It also demonstrates that single 5′UTR sites with more physiologic partial base pairing to miR-10b confer weak/moderate regulation (in the range of 20–50%), and the correspondingly weak effects on the expression of encoded proteins. Therefore, the strong effect of miR-10b inhibition on glioma and GSC viability is a cumulative result of derepression of numerous genes, regulated through the seed and other miR-10b binding sites in various mRNA regions, with a preferential binding via 5′UTRs. The factors and molecular mechanisms favoring miR-10b binding to 5′UTRs are currently unknown; they might include specific RNA-binding proteins interacting with miR-10b as well as regulatory regions in 5′UTR, or translational machinery directly. Further investigation of such unconventional mechanisms will not only highlight the additional layers of gene regulation, but may also explain the strong context specificity of miR-10b function observed, for example, between the brain and breast cancers (Ma *et al*, 2007, 2010; Gabriely *et al*, 2011a).

Another objective of this study was to further characterize miR-10b as a potential therapeutic target for GBM and explore various strategies of miR-10b inhibition in orthotopic GBM. Of note, common downstream targets modulated in heterogeneous human GSC cultures and glioma lines by miR-10b ASO and identified in this study provide critical biomarkers/readouts for the efficacy of miR-10b inhibition *in vivo*. Previous experiments on subcutaneous U87 glioma suggested that miR-10b inhibition could reduce tumor growth (Gabriely *et al*, 2011b). In addition, *ex vitro* silencing of miR-10b in U87-derived mesenchymal glioma cell line reduced the growth of orthotopic tumors derived from these cells (Lin *et al*, 2012). Here, we demonstrated for the first time that delivery of miRNA inhibitors to intracranial GBM impairs the growth of these highly aggressive tumors and provides survival benefits. We employed two distinct orthotopic GBM models, one based on human GSC (GBM8) xenografts that form diffusely invasive brain tumors in nude mice (Wakimoto *et al*, 2012), while another based on mouse GL261 glioma allografts in syngeneic immunocompetent mice. Furthermore,

we examined various formulations and delivery modalities for the miR-10b ASO inhibitors. Whereas ASO-based drugs, including miRNA inhibitors, are actively pursued for various therapeutic applications, their investigation and use for brain and CNS disorders have been limited due to a number of challenges, mostly associated with intracranial delivery. In the initial set of experiments, we utilized local intratumoral injections of miR-10b ASO and observed the derepression of targeted splicing factors and significant attenuation of GBM growth. Although this strategy supported the therapeutic potential of miR-10b inhibition for GBM, it required repeated neurosurgery and might not be applicable for human patients. Therefore, we also utilized osmotic delivery that provides continuous, local, and targeted intratumoral infusion of the ASO to intracranial GBM, requires a single surgical procedure, and may represent efficient therapeutic approach for malignant brain tumors. Importantly, glioma cells are highly proliferative, and substantial levels of miR-10b expression in these cells are ensured by its active transcription. Continuous delivery of the ASO inhibitor can improve its bioavailability and help overcome dilution in an enlarging GBM. A similar approach has been successfully utilized in a recent clinical trial with antisense inhibitors of TGFβ (Bogdahn *et al*, 2011). Our experiments with lipid nanoparticles loaded with miR-10b ASO (at the low daily dose of 2 μg) demonstrated that this approach reduced GBM growth in both athymic and immunocompetent mice. However, it was not sufficient to fully eradicate the growth of the tumors. Since miR-10b inhibition in cultures ultimately causes death of all glioma cells and cell types, with no innate or acquired resistance observed, our data suggest that additional optimization of ASO nanoparticle dosage and formulations, and improvement of their delivery, may translate to high *in vivo* efficacy.

One of the most significant aspects of this study is associated with promising data provided by the systemic administration of miR-10b ASO. Generally, systemically injected ASO that contain nuclease-resistant phosphorothioate backbone distribute broadly into most tissues except CNS (Geary *et al*, 2015). Both size and chemistry prevent their delivery and distribution across the blood–brain barrier (BBB). GBM, however, is characterized by disruption of the BBB, suggesting a possibility of systemic treatments based on such ASO. Here, we utilized uncomplexed 2′-O-MOE-phosphorothioate miR-10b ASO for i.v. injections (at 80 mg/kg) and found it to be effective for inhibiting established intracranial GBM8. To the best of our knowledge, this work offers the first indication that systemically administered antagomirs can be taken up by glioblastoma, inhibit a miRNA, and lead to target derepression in intracranial gliomas. Notably, only a small portion of the ASO has been delivered to GBM; nevertheless, it resulted in twofold downregulation of miR-10b in the tumor tissues, and small but significant derepression of its mRNA targets. It was sufficient to delay GBM progression, did not cause toxicity, and was well tolerated. Furthermore, daily s.c. administration of high-dose miR-10b ASO (at 100 mg/kg) over 30 days proved safe in mice, suggesting that systemic miR-10b inhibition can be tolerated well clinically, despite high miR-10b levels in various normal extracranial tissues and cells. Of note, miR-10b knockout mice do not exhibit a pathological phenotype (Park *et al*, 2012), http://jaxmice.jax.org/strain/016950.html, suggesting that miR-10b activity is dispensable for normal cells, at least in mice, and further supporting the systemic inhibition approach for GBM patients.

In conclusion, in this work we investigated the common mechanism underlying miR-10b activity in GSC and further validated miR-10b as a promising therapeutic target. We presented next steps toward miR-10b-based therapy development for GBM and identified common mRNA targets that could be utilized as therapeutic readouts. Various orthotopic models of GBM, including human GSC xenografts in immunocompromised mice, and chemically induced mouse glioma allografts in immunocompetent mice, are highly responsive to anti-miR-10b therapy. Different formulations of miR-10b inhibitors and routes of administration explored in this study provide rationale for more detailed investigation that has to focus on the optimization of ASO chemistry, formulation, dosage, and delivery for clinical applications.

# Materials and Methods

### Cell cultures and transfections

Human low-passage GBM stem-like cells (GBM4, GBM6, GBM8, and BT74) were a generous gift from Dr. Hiroaki Wakimoto. The tumorigenic, genetic, and stem properties of these cells have been previously described (Wakimoto *et al*, 2009, 2012). The cells were maintained in serum-free neurosphere cultures in Neurobasal media supplemented with 1× B27 and 0.5× N2 (Invitrogen), 3 mM L-glutamine, 50 units/ml of penicillin, 50 units/ml of streptomycin, 125 ng/ml of amphotericin B (Cellgro), 2 μg/ml of heparin (Sigma-Aldrich), 20 ng/ml of FGF2 (PeproTech), and 20 ng/ml of EGF (R&D systems). The cells were passaged by dissociation using Neurocult Stem Cells chemical dissociation kit (Stem Cells Technologies). Mouse chemically induced GBM cell line GL261 was obtained from Dr. Bozena Kaminska and maintained in monolayer cultures in DMEM with 10% FBS. GL261 cells were passaged using Accumax Cell Dissociation reagent (Innovative Cells Technologies). Human glioma cell lines and breast epithelial MCF7 line were obtained from American Type Culture Collection (ATCC), cultured in DMEM/10% FBS (Gibco), and passaged by trypsinization. miRNA inhibitors and mimics were transfected at 50 nM final concentration, using Lipofectamine 2000 and Oligofectamine (Invitrogen), respectively. For transfections of GSC, the neurospheres were dissociated to single cell suspension prior to addition of the transfection mix. Fluorescently labeled non-targeting oligonucleotides of matching chemistries were used to monitor transfection efficiencies. All cell lines were tested for mycoplasma contamination before use.

### miRNA inhibitors and mimics

Antisense oligonucleotide inhibitors with nucleoside sugar modifications (2′-O-methoxyethyl), with phosphodiester (PO), phosphorothioate, and mixed PS/PO backbones, were synthesized and generously provided by Regulus Therapeutics, Inc., San Diego, CA. The oligonucleotide sequences were as following: 5′-CACAAATTCGGTTCTACAGGGTA-3′ (miR-10b ASO) and 5′-ACATACTCCTTTCTCAGAGTCCA-3′ (non-targeting control of the same chemistry). The efficiency and specificity of miR-10b inhibitors used in this study have been validated previously (Gabriely *et al*, 2011b). Cy5-labeled non-targeting 2′-O-MOE-PS oligonucleotide was used to monitor the systemic delivery to the orthotopic tumor *in vivo*. miR-

10b mimic and matching double-stranded RNA control oligonu-cleotide were obtained from Ambion.

## The whole-genome expression profiling by Affymetrix microarrays and data analysis

GBM4, GBM6, and GBM8 cells were transfected in duplicates with either miR-10b inhibitor or non-targeting control oligonucleotide, or treated with Lipofectamine 2000 alone ("mock"). The cells were harvested 24 h after transfections, total RNA was isolated using TRIzol reagent (Invitrogen), and RNA integrity was examined by Agilent Bioanalyzer. Whole-genome microarray expression analysis was conducted using a commercial array (Hgu133plus2, Affymetrix). Microarray data were background-adjusted, quantile-normalized, and summarized using the GC Robust Multiarray Average (GCRMA) method using R (www.r-project.org) and the Bioconductor (www.bioconductor.org) package *gcrma*. Differential expression analysis between miR-10b inhibitor versus non-targeting control or mock-treated samples was done using the Bioconductor package *limma* for R (www.bioconductor.org). Pathway analyses were performed using IPA (Ingenuity Systems, http://www.ingenuity.com) and GSEA (www.broadinstitute.org/gsea/). The heatmaps of differentially expressed genes ($0.8 >$ fold change $> 1.2$, and Student's $t$-test $P \leq 0.05$ in at least two cell lines) were generated with the heatmap.2 function of the *gplots* package in R (Version 3.0.3).

For identification of miR-10b-binding motifs, based on the arrays datasets, we utilized three groups of transcripts that were up-regulated (with fold changes higher than 1.2), down-regulated (with fold changes lower than 0.8), and unchanged (with fold changes between 0.8 and 1.2) in response to miR-10b ASO. Using Ensembl annotation (GRCh37), we explored the content of 3′UTR, CDS, and 5′UTR fragments of the transcripts using all subsequences of the mature miR-10b sequence with the motif length ranging from 6 to 9 bp. We calculated frequencies of the miR-10b-binding motifs in 3′UTRs, CDS, and 5′UTRs of up-regulated, down-regulated, and unchanged transcripts. The calculated values were normalized by the number of up-regulated, down-regulated, and unregulated transcripts. For each motif and transcript region, we identified the ratios of normalized frequencies in up-regulated versus down-regulated, and in up-regulated versus unchanged transcripts. We repeated the same procedure for 1,000 randomly shuffled sequences of mature miR-10b and calculated the probabilities that inspected ratios were higher than 1.5 not by chance in the cases of mature miR-10b subsequences.

## Analysis of gene expression by quantitative real-time reverse transcriptase PCR (qRT–PCR)

Analysis of gene expression by qRT–PCR was performed as previously described (Teplyuk *et al*, 2015). Primers sequences are listed in Appendix Table S2. For miRNA expression analysis, 3.3 ng of total RNA was used in reverse transcription reaction, followed by qRT–PCR using TaqMan MicroRNA Assays (Life Technologies). The fold change in expression was calculated by $\Delta C_t$ method.

## Western blot analysis

Western blot analysis was performed by standard procedure, as previously reported (Gabriely *et al*, 2011b). The following primary

antibodies were used: cleaved caspase 3 and cleaved caspase 7 (Cell Signaling, #9661 and #9491, dilution 1:1,000), beta-actin (Abcam, ab3280, dilution 1:5,000), MBNL1 (Abcam, ab108519, dilution 1:80), MBNL3 (Sigma, SAB1411751, dilution 1:50), SART3 (Sigma, SAB2104147, dilution 1:500), and RSRC1 (Sigma, SAB3500150, dilution 1:2,000). The anti-MBNL2 mouse antibody was generously provided by Dr. Glenn Morris (Holt *et al*, 2009) and used in dilution 1:100. Secondary antibodies from Cell Signaling diluted at 1:1,000 were used. Antibodypedia (http://www.antibodypedia.com) and Degreebio (http://1degreebio.org) resources were used for validation profile of the antibodies.

## Analysis of GSC neurospheres

GSC were dissociated to single cell suspension, plated in 6-well plates at $0.5 \times 10^6$ cells per plate, and transfected with miR-10b ASO as described. The neurospheres were collected five days after transfection, diluted sixfold with fresh media, and transferred to 96-well tissue culture assay plates, 100 μl per well. To examine cell viability, metabolic activity of the cells was measured using CellTiter-Glo Luminescent Cell Viability Assay (Promega). To assess the number and size of the neurospheres, the pictures were taken and the analysis performed using the ImageJ software, with average values calculated for five wells per experimental condition. For the analysis of apoptosis, the cells were harvested at day 5 after transfection, washed with PBS, and stained with antibodies to Annexin and propidium iodide using Annexin-V-FLUOS Staining Kit (Roche). The cells were spun on microscopic slides and mounted, and fluorescence microscopy was conducted within 2 h after staining. Alternatively, neurospheres were double-stained with Annexin V/7-AAD, dissociated, and analyzed by flow cytometry.

## GSC differentiation

To induce GSC differentiation, cells were dissociated and plated on polyornithine and fibronectin double-coated plates in differentiation media (Neurobasal media supplemented with B27, N2, 3 mM L-glutamine, and 5% FBS), as described (Wakimoto *et al*, 2009).

## Reporter constructs and luciferase reporter assays

Full-length 5′UTRs and 3′UTRs of candidate target genes were amplified from A172 cDNA and cloned into pLightSwitch 5′UTR reporter (Active Motif) or PsiCheck2 3′UTR reporter (Promega), respectively. Primers for amplification are listed in Appendix Table S3. The 5′UTR constructs with deleted miR-10b binding sites were generated using Q5 site-directed mutagenesis kit (NEB). 5′UTR miR-10b reporter was produced by inserting miR-10b complementary sequence between BglII and NcoI sites in the 5′UTR of pLightSwitch_5UTR vector. For the reporter assays, MCF7 cells expressing low levels of endogenous miR-10b were plated in 96-well plates and sequentially transfected with miR-10b mimic/control, followed by the transfection with 40 ng/well of reporter plasmids 24 h later. Luciferase activity was measured 24 h after the second set of transfections with Dual-Glo Luciferase Assay System (Promega E2920). Six wells were quantified per condition.

## Stereotaxic injection of glioma cells and whole-body imaging (WBI)

To enable the monitoring of orthotopic GBM in mice, human GSC (GBM8) and mouse GL261 glioma cells have been transduced with lentiviral vector CSCW2-Fluc-ImCherry as previously described (Maguire *et al*, 2008). Tumor cells ($1 \times 10^5$) or small spheres expressing *firefly* luciferase and mCherry were implanted into the striatal area (coordinates 0.5 mm RC, 2.0 mm LL, and -2.5 mm DV from lambda) of 6-week-old athymic (nude) female mice or male Black 6 Albino mice (Charles River Laboratories). The injections were carried out using digital stereotaxic frame instrument (Stoelting) equipped with UMP3 pump and pump controller. Tumor growth was monitored every 3–4 days after cell implantation by luciferase bioluminescence imaging as described (Teng *et al*, 2014). Data acquisition, processing, quantification, and visualization were carried out using Living Image 4.2 software.

## Treatment of intracranial GBM tumors with miR-10b inhibitors

The treatments have been initiated when bioluminescence reached the exponential phase, reflecting active tumor growth. The mice were randomized to the "treatment" and "control" groups based on the WBI, with similar average bioluminescence signal and tumor growth rates per group. For intratumoral injections, miR-10b 2′-O-MOE-PO ASO or non-targeting oligonucleotide was formulated with *in vivo* jetPEI transfection reagent (Polyplus), and 1 μg of the oligonucleotides was injected in 2 μl by stereotaxic brain surgery at days 20 and 25 after cell implantation. Tumor growth rates were monitored by WBI, and mice were sacrificed at day 34. The brains were fresh-frozen, and tumors were dissected for RNA analysis. For systemic delivery, uncomplexed stabilized 2′-O-MOE-PS miR-10b ASO or the corresponding non-targeting control was utilized. In the case of GBM8 tumors, 80 mg/kg ASO was injected through the tail vein at days 11, 21, 25, 27, and 31 after cell implantation. The mice were sacrificed at day 34 after cells implantation, and the brains were fresh-frozen and sectioned for histology, with a part of each tumor dissected for RNA analysis. For GL261 tumors, 100 mg/kg of each oligonucleotide was administered subcutaneously daily until the endpoint of the experiment.

For continuous osmotic delivery, ALZET® Osmotic infusion pumps (model 2002) were loaded with 0.17 μg/μl of 2′-O-MOE-PO miR-10b ASO or control oligo, formulated with cationic lipid nanoparticles. Lipid nanoparticles were prepared by a modified ethanol injection method as described previously (Batzri & Korn, 1973; Jaafar-Maalej *et al*, 2010). Cationic lipid, cholesterol, phospholipid, and Pegylated lipid were dissolved in ethanol. Anti-miR was dissolved in an aqueous buffer. The lipid mixture was added by means of a syringe pump to the anti-miR solution under stirring. Lipid nanoparticles thus obtained were diluted with phosphate-buffered saline (pH 7.4) followed by purification using tangential flow filtration. The purified lipid nanoparticles were concentrated by diafiltration and subjected to sterile filtration through a 0.8/0.2-μm syringe filters into sterile vials and stored at 2-8°C. Characterization of the lipid nanoparticles involved measuring particle size, polydispersity, total oligonucleotide anti-miR content, and percentage of free oligonucleotide. Particle size and polydispersity were measured on a Malvern zetasizer. Total anti-miR content of the nanoparticles was determined using HPLC, while percentage of free anti-miR was determined using a ribogreen assay.

The pumps and a micro-infusion cannula (Alzet brain infusion kit 3) for controlled delivery of antagomirs were stereotactically inserted at the same coordinates at day 26 after injections of GBM8 cells. About 2 μg of inhibitors was infused daily for the duration of 14 days. Tumor growth was monitored and animals sacrificed at the end of treatment. Brains were fixed in 4% formaldehyde, frozen, and cryosectioned, and immunostains were performed as described. Osmotic delivery to GL261 tumors has been carried out using similar pumps (model 2001), with the following modifications. The pumps were loaded with 2′-O-MOE-PS/PO miR-10b ASO or the corresponding control oligonucleotide formulated with *in vivo* jetPEI reagent, and implanted to Black 6 Albino mice on day 6 after GL261 cell injections. About 2 μg of inhibitor was infused daily for the duration of 7 days. The brains were sectioned for immunostaining, with a part of each tumor dissected for RNA analysis. All animal studies, including animal survival experiments, have been approved by the Harvard Medical Area (HMA) Standing Committee on Animals and were conducted in compliance with ARRIVE guidelines (Kilkenny *et al*, 2010), as well as NIH (http://grants.nih.gov/grants/olaw/olaw.htm) and MRC (http://www.mrc.ac.uk/about/policy/policy-highlights/research-involving-animals/) recommendations.

## Immunohistochemistry

Intracranial tumors were fixed with 4% formaldehyde, embedded, frozen, and sectioned according to standard procedures. Staining of 10-μm-thick sections was performed using PCNA (ab18197, Abcam, 1:500 dilution), Ki67 (ab15580, 1:200), and cleaved caspase 3 (ab32042, 1:200) antibodies. Rabbit polyclonal antibodies recognizing oligonucleotides with phosphorothioate backbone were generously provided by Regulus Therapeutics, Inc., and used at 1:200 dilution. Hematoxylin (Gill 2) and eosin (Sigma) staining was performed according to standard procedures.

## Statistical analysis

The differences between groups were analyzed using two-tailed unpaired Student's *t*-test. The adequate samples sizes were calculated based on Resource equation method (Festing & Altman, 2002). The animals were randomized to the "treatment" and "control" groups based on the WBI, with similar average bioluminescence signal and tumor growth rates per group. Normality test was used to assess data distribution. No blinding of investigator was used in the experiments. Survival was analyzed by log-rank test using a commercial software (Medcalc).

## Data availability

The microarrays data have been deposited in NCBI's Gene Expression Omnibus (Edgar *et al*, 2002) and are accessible through GEO Series accession number GSE68424 (http://www.ncbi.nlm.nih.gov/geo/query/acc.cgi?acc = GSE68424).

**The paper explained**

**Problem**

Glioblastoma (GBM) is the most aggressive primary brain tumor in adults. The mean survival is fourteen months after initial diagnosis and surgical resection. Few drugs were found to be effective for GBM, and even those offer only marginal improvement of progression-free survival or overall survival. GBM stem-like cells (GSC) are considered essential for tumor growth by being the sole source of progeny tumor cells. GSC were found to be highly resistant to current therapies. There is an urgent need for a more efficient treatment and this would likely include better targeting of GSC.

**Results**

In this work, an oncogenic microRNA-10b (miR-10b), highly abundant across all GBM subtypes, was characterized as a potent therapeutic target that is critical for the viability of GSC. We identified several splicing factors as common downstream targets regulated by miR-10b in GSC, and potential biomarkers for miR-10b inhibition *in vivo*. Inhibition of miR-10b with specific antisense oligonucleotides (ASO) disrupts cell cycle progression and viability of GSC, whereas it does not affect normal neurons or astrocytes, owing to absent miR-10b. Treatment of established intracranial GBM with miR-10b ASO slows down tumor growth and prolongs animal survival in two different models of the disease: human GSC-derived tumors in nude mice and murine chemically induced GL261 tumors in immunocompetent mice. Three delivery routes for miR-10b ASO (intratumoral injections, continuous intratumoral delivery by osmotic pumps, and systemic administration) were tested and demonstrated efficacy and safety.

**Impact**

This work addresses an unmet clinical need and demonstrates the feasibility of therapeutic miRNA targeting for GBM. The obtained data has a high translational impact as it promotes a new and safe strategy for GBM treatment, based on local or systemic miR-10b inhibition, and potentially efficacious for all subtypes of GBM.

Expanded View for this article is available online.

## Acknowledgements

This work was supported by grants from NIH/NCI (RO1CA138734), Sontag Foundation Distinguished Scientist Award, and National Brain Tumor Society (to AMK), and Brain Science Foundation (to NT, AMK, and EJU). We thank Anant Jain, Deeptha Vasudevan, and Meenakshi Basu for technical assistance, Dr. Oleg Butovsky for assistance with IPA analysis, and Dr. Vivek Kaimal for assistance with microarray analysis. We thank Regulus Therapeutics, Inc., for providing miRNA inhibitors and antibody recognizing the ASO. We also thank Dr. Hiroaki Wakamoto for providing human GBM stem cells, Dr. Bozena Kaminska for mouse GL261 cells, and Dr. Glenn Morris for sharing the anti-MBNL2 antibody.

## Author contributions

NMT developed project directions, conceived, designed, and performed majority of the experiments, acquired and analyzed results, coordinated work of co-authors, and wrote the manuscript. AMK originated, conceived, and supervised the project and wrote the manuscript. EJU, GG, BT, JT, MP, AM, and YK assisted with experiments and contributed to the data analysis. PK and EM designed and synthesized ASO and created lipid nanoparticles formulations. NV, YW, and RC performed bioinformatics analysis. EAC, JG, and EJU helped with data analysis and critically revised the manuscript. All authors approved the manuscript.

## Conflict of interest

Priya Karmali and Eric Marcusson were employees of Regulus Therapeutics.

## For more information

TCGA data portal: https://tcga-data.nci.nih.gov/tcga/
RNA hybrid: http://bibiserv.techfak.uni-bielefeld.de/rnahybrid/
Expression Microarray data deposited to GEO: http://www.ncbi.nlm.nih.gov/geo/query/ acc.cgi?acc=GSE68424

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
