## [Review Process File · EMBO Molecular Medicine]

Manuscript EMM-2015-05495

Therapeutic potential of targeting microRNA-10b in established intracranial glioblastoma: first steps toward the clinic.

Nadiya M. Teplyuk, Erik J. Uhlmann, Galina Gabriely, Natalia Volfovsky, Yang Wang, Jian Teng, Priya Karmali, Eric Marcusson, Merlene Peter, Athul Mohan, Yevgenya Kravtsov, Ron Cialic, E. Antonio Chiocca, Jakub Godlewski, Bakhos Tannous, Anna M. Krichevsky.

Corresponding author: Anna Krichevsky, Brigham and Women's Hospital and Harvard Medical School

Review timeline:

Submission date:	01 June 2015
Editorial Decision:	06 July 2015
Revision received:	23 November 2015
Editorial Decision:	21 December 2015
Revision received:	22 December 2015
Accepted:	11 January 2016

Transaction Report:

Editor: Céline Carret

1st Editorial Decision

06 July 2015

Thank you for the submission of your manuscript to EMBO Molecular Medicine. I am very sorry for the long time it took us to get back to you. In this case we experienced difficulties in securing three expert and willing Reviewers, we had to compromise for two. They now have evaluated your manuscript and the reports are pasted below.

As you will see, both referees find the topic very important and the prospects of a new therapeutic of great interest. However, they both suggest strengthening the paper by 1) experimentally addressing the translational implication of the data by testing different routes and means of delivery, and 2) improving the underlying mechanism (referee 1). In addition, clarifications and details should be provided as requested. However, while referee 2 suggests separating mechanistic from translational insights, given our journal's scope, I would advice against and keep both aspects in, only further developing them.

We feel that the suggested revisions are reasonable and would considerably improve the manuscript, therefore, we would encourage you to address all issues as best as possible. Please note that it is EMBO Molecular Medicine policy to allow only a single round of major revision and that, as acceptance or rejection of the manuscript will depend on another round of review, your responses

should be as complete as possible.

Please read below for important editorial formatting.

I look forward to receiving your revised manuscript.

***** Reviewer's comments *****

Referee #1 (Remarks):

GBM is a very lethal tumor and it is paramount to look for new therapies. In the manuscript entitled "Therapeutic potential of targeting microRNA-10b in established intracranial glioblastoma: first steps toward the clinic." Teplyuk and colleagues deal with an extremely interesting and promising topic whose future developments could pave the way to therapeutic implications. In 2011, the same authors published a paper on Cancer Research entitled "Human glioma growth is controlled by microRNA-10b" highlighting the involvement of miR-10b in the regulation of cell cycle progression in several GBM cell lines, both in vitro and in vivo. Now they show the role of miR-10b in GBM initiating stem-like cells (GSC) in vitro and in vivo. They show very convincing pre-clinical results based on the miR-10b therapy in GBM. In vivo results are well thought and elegantly performed; they have used multiple approaches to demonstrate the utility of treatment with miR-10b inhibitors. Some modifications are needed to render the manuscript more convincing.

Major Points:

- Since the novelty of the manuscript is the study of the role of miR-10b in GSC, it is worthy that the authors explore the effects of miR-10b treatment on GSCs properties. It has been already demonstrated by the authors that miR-10b is able to influence cell cycle in non-GSC, but how is it relevant in cancer stem cell setting? Does it alter differentiation, self-renewal, tumor-initiation capacity?
- I found the therapeutic exploitation of miR-10b of great interest. The authors should show the inhibition of miR-10b in the GL261 glioma model also by systemic injections; this would address concerns about possible side effects in the allograft model and would strengthen the putative druggability of miR-10b.

Minor Points:

- In Figure 1B the authors claim that miR-10b inhibition results in fewer neurospheres. To avoid possible bias due to sphere fusion the assay must be performed in semi-solid substrates (i.e methylcellulose).
- In Figure 1C and D the authors should quantify PI/Annexin V and cleaved-caspase 7 by FACS.
- I don't see how Figure 4E would support author conclusions. Out of five splicing regulators, only two of them (MBNL2 and MBNL3) support the trend highlighted by the authors while MBNL1, SART3 and RSRC1 move in different directions.
- In Figure 5F the values of tumor growth of control mice are highly dispersed, even if at a quick glance they may seem different, three points out of five fall in the 0-300 range, the same as miR-10b-i. Could the authors perform a t-test showing statistical significance?
- Why miR-125b has been used as normalizer in Figure 5B?
- Why miR-16 has been used as normalizer in Figure Supplementary 1? (In Ref.7 the authors

demonstrated that miR-16 activity is modified by miR-10b...)

Referee #2 (Comments on Novelty/Model System):

for in vivo experiments additional lines should be used as suggested

Referee #2 (Remarks):

Review Teplyuk et al

Mir10b has been described as an oncomir in several tumor types including glioma. The manuscript details the attempt to identify cancer relevant targets of miR10b using differential gene expression profiling of sphere lines using a knock down approach in order to understand the underlying mechanism resulting in the pro-oncogenic role. In addition the development of a treatment strategy targeting mir10b is presented. Although of potential high scientific and clinical interest, due to the limitations of the experimental design and choice of procedures (bioinformatics; use of only one line in vivo), the paper is neither strong for advancing the mechanistic insights, nor the therapeutic approach. It would be immature to publish this work at the present stage. The authors may also consider separating the 2 topics, as at present there is limited input from the mechanistic side to the preclinical aspect, testing different routes and means of delivering the drug.

Major comments

Differentially expressed genes:

Using differential gene expression profiling of 3 GBM sphere lines with or without knock-down of miR10b gene lists are derived with up- or downregulated genes.

- The list is statistically of little value, as there are only few lines, without biological replicates, no multiple testing correction, and the difference to be observed is a fold change of 1.2 (log space?, no indication in the text or methods), in two of three lines at $p < 0.5$. This could be corrected if a strong functional validation of genes of interest would follow.

The genes are annotated by gene ontology and described as being enriched for cell cycle genes, which was known before and having functions in RNA processing and RNA splicing. Given the latter is not easy to functionally validate they check if a selected set of genes is in general more highly or more lowly expressed than in normal brain in publically available datasets (Fig3). This is not really pertinent!

- These analyses should be replaced by the correlation of the genes of interest with miR10b in the TCGA GBM and LGG data-sets where miR profiles are available. On this occasion the TCGA data could also be used to determine if the genes identified in the screen were the top correlated genes, or whether there are other gene sets that may be more relevant (statistically, biologically).

Figure S4 claims to identify alternatively spliced genes due to knock-down of miR10b, I don't think that the AFFYmetrix chip used is an appropriate tool to detect alternative splicing. At least it would call for respective validation.

- The genes followed up on that may be modulated by miR10b using reporter experiments would profit from validation in the TCGA data as suggested above.

- None of the genes is followed up on for biological evaluation and relevance for the observed phenotype.

The in vivo experiments could profit from using several lines with distinct genotypes, or for the matter of testing migration using a truly invasive model. Figure 8 suggests that the model used is not really invasive.

Minor comments

- There are references in the text to tables in the supplement with gene lists. The tables have no numbers or title, and it is not clear what they comprise. At present they are not very helpful.

1st Revision - authors' response

23 November 2015

Referee #1

GBM is a very lethal tumor and it is paramount to look for new therapies.

In the manuscript entitled "Therapeutic potential of targeting microRNA-10b in established intracranial glioblastoma: first steps toward the clinic." Teplyuk and colleagues deal with an extremely interesting and promising topic whose future developments could pave the way to therapeutic implications. In 2011, the same authors published a paper on Cancer Research entitled "Human glioma growth is controlled by microRNA-10b" highlighting the involvement of miR-10b in the regulation of cell cycle progression in several GBM cell lines, both in vitro and in vivo. Now they show the role of miR-10b in GBM initiating stem-like cells (GSC) in vitro and in vivo. They show very convincing pre-clinical results based on the miR-10b therapy in GBM. In vivo results are well thought and elegantly performed; they have used multiple approaches to demonstrate the utility of treatment with miR-10b inhibitors.

Some modifications are needed to render the manuscript more convincing.

Major Points:

1. Since the novelty of the manuscript is the study of the role of miR-10b in GSC, it is worthy that the authors explore the effects of miR-10b treatment on GSCs properties. It has been already demonstrated by the authors that miR-10b is able to influence cell cycle in non-GSC, but how is it relevant in cancer stem cell setting? Does it alter differentiation, self-renewal, tumor-initiation capacity?

Response: To address the question whether miR-10b affects self-renewal and differentiation capacity of GSCs, we cultured GBM4 and GBM8 cells in commonly used differentiating conditions (as attached cultures in the media containing 5% FBS, and deprived of growth factors [1]), with or without miR-10b inhibition. The expression of markers of GSCs stemness/self-renewal (Nestin, OCT4) and differentiation (GFAP) were measured by qRT-PCR during differentiation. We found that inhibition of miR-10b attenuates expression of stemness markers (Nestin and OCT4), while increases expression of astrocytic marker GFAP, therefore prompting cells towards astrocytic differentiation. The results of these new experiments are included in Results section of the manuscript (page 5) and in the new Appendix Figure S2.

2. I found the therapeutic exploitation of miR-10b of great interest. The authors should show the inhibition of miR-10b in the GL261 glioma model also by systemic injections; this would address concerns about possible side effects in the allograft model and would strengthen the putative druggability of miR-10b.

Response: To explore the effect of systemically delivered miR-10b inhibitor in GL261 allograft model; we first conducted an experiment to study the efficiency of systemic anti-miR-10b delivery to intracranial GL261 tumors. This experiment was necessary since GL261 tumors are significantly less invasive and diffusive, and may have less "leaky" blood brain barrier than GBM8 xenografts (as well as human GBM), for which systemic way of delivery was efficient. The tumors were developed by implantation of GL261 cells into the brain of BLACK6 syngeneic mice, and anti-miR-10b ASO inhibitor (100 mg/kg daily) was administered systemically through both IV (via the tail vein) and subcutaneous injections. Both types of systemic injections resulted in apparent delivery of miR-10b inhibitor to growing GL261 tumors. However, the efficiency of systemic delivery to GL261

tumors was significantly lower than to GBM8 tumors, with only around 10% of GL261 cells positive for oligonucleotide presence, in comparison with 70-80% of positive GBM8 cells. We have concluded that there is substantial delivery to GL261 tumors, however, of lower efficacy than to GBM8, most likely because of more compact structure and lower invasiveness of GL261 tumors.

We have further studied the effect of systemic treatment of GL261 tumors with miR-10b inhibitor. Both average tumor growth rate and mouse survival were slightly affected by miR-10b inhibitor relative to non-specific control, which was expected considering the low uptake of ASO by GL261 glioma. Importantly, we have not observed any toxicity of systemic miR-10b inhibition, at the high dose used (100 mg/kg daily) over the prolonged course of treatment (14 days). Therefore, even though this model does not recapitulate the invasive character of human GBM and may not be optimally suited to evaluate the therapeutic efficacy of systemic anti-miR-10b delivery to human GBM, this new set of experiments demonstrates safety of systemic miR-10b inhibition. We believe these results are highly important; they are presented in Figure EV5 and described on page 11 of the revised manuscript.

Minor Points:

1. In Figure 1B the authors claim that miR-10b inhibition results in fewer neurospheres. To avoid possible bias due to sphere fusion the assay must be performed in semi-solid substrates (i.e methylcellulose).

Response: The reason for decrease in the number of neurospheres after anti-miR-10b treatment is not sphere fusion, since not only the number but also the diameter of the spheres is strongly reduced (Fig. 1B). We now modified the text to better describe the observed effects that were sequence-specific, dramatic, and highly reproducible (page 5), as follows:

Consistent with previous data, inhibition of miR-10b had a strong effect on viability of GSC (Figure 1A). GSC transfected with miR-10b ASO formed neurospheres similar to control cultures, suggesting that neurosphere-forming capacity was unaltered. However, at later time point (day four post-transfection) the neurospheres started to exhibit the markers of apoptosis followed by the massive cell death and sphere disaggregation. This process resulted in significant reduction of both number and size of the GSC neurospheres (Figure 1B).

2. In Figure 1C and D the authors should quantify PI/Annexin V and cleaved-caspase 7 by FACS.

Response: As suggested by the reviewer, we have quantified the amount of apoptotic GBM8 cells by flow cytometry analysis. We found that the numbers of both 7-Amino-Actinomycin and Annexin V positive cells were significantly higher in cells treated with miR-10b inhibitor relative to the control cells at day four after transfection, similarly to what we observed previously by fluorescence microscopy. The results of flow cytometry analysis are included as new Figure 1D panel, while microscopy images of Propidium Iodide and Annexin V staining are moved to Figure EV2.

3. I don't see how Figure 4E would support author conclusions. Out of five splicing regulators, only two of them (MBNL2 and MBNL3) support the trend highlighted by the authors while MBNL1, SART3 and RSRC1 move in different directions.

Response: The results of the Luciferase reporter assays indeed indicate that only MBNL2 and MBNL3 5'UTR constructs are directly regulated by miR-10b binding within the proposed motifs, as their expression is fully rescued by the mutations within the predicted miR-10b binding sites. The MBNL1 5' UTR construct did not consistently respond to changes in miR-10b levels, and miR-10b

regulation of *SART3* and *RSRC1* constructs was incompletely rescued by mutations of the predicted miR-10b binding sites. Therefore, this artificial assay has not provided firm validation of three late factors as direct miR-10b targets. Nevertheless, regulation of their mRNAs and proteins by both miR-10b inhibitors and mimics in multiple glioma cell lines (Fig. 4B,C), and strong putative binding sites present in the UTRs, are highly suggestive of direct regulation. To make this point clear, we now introduced the corresponding text at the page 5 of the manuscript.

4. In Figure 5F the values of tumor growth of control mice are highly dispersed, even if at a quick glance they may seem different, three points out of five fall in the 0-300 range, the same as miR-10b-i. Could the authors perform a t-test showing statistical significance?

Response: T-test of the results shown in Figure 5F indicates a statistical significance of difference between control and treatment groups at the end of the experiment (day 29 of treatment), with p value = 0.023277. We have indicated the difference by an asterisk in Figure 5F.

5. Why miR-125b has been used as normalizer in Figure 5B?

Response: miR-125b was used to normalize miR-10b expression since its expression level was stable between the samples and not affected by miR-10b inhibition. Several other normalization strategies have not altered the results.

6. Why miR-16 has been used as normalizer in Figure Supplementary 1? (In Ref.7 the authors demonstrated that miR-16 activity is modified by miR-10b...)

Response: We appreciate the Reviewer's comment and now show the normalization to "housekeeping" snoRNA U6, which levels are stable in the normal brain and GBM (Figure EV1). Of note, the principal conclusion that miR-10b is not expressed in the normal fetal and adult brain and normal neuroglial cells but highly expressed in GBM, GBM lines and GSC cultures, is not altered by various normalization strategies we have utilized.

Referee #2:

Mir10b has been described as an oncomir in several tumor types including glioma. The manuscript details the attempt to identify cancer relevant targets of miR10b using differential gene expression profiling of sphere lines using a knock down approach in order to understand the underlying mechanism resulting in the pro-oncogenic role. In addition the development of a treatment strategy targeting mir10b is presented. Although of potential high scientific and clinical interest, due to the limitations of the experimental design and choice of procedures (bioinformatics; use of only one line in vivo), the paper is neither strong for advancing the mechanistic insights, nor the therapeutic approach. It would be immature to publish this work at the present stage. The authors may also consider separating the 2 topics, as at present there is limited input from the mechanistic side to the preclinical aspect, testing different routes and means of delivering the drug.

Response: We appreciate the Reviewer's comments and critiques and believe that, by addressing them, we significantly improved the manuscript, as outlined below.

Major comments

1. Differentially expressed genes:

Using differential gene expression profiling of 3 GBM sphere lines with or without knockdown of miR10b gene lists are derived with up- or down regulated genes. The list is statistically of little value, as there are only few lines, without biological replicates, no multiple testing correction, and the difference to be observed is a fold change of 1.2 (log space? no indication in the text or methods), in two of three lines at $p < 0.5$. This could be corrected if a strong functional validation of genes of interest would follow.

Response: The microarray expression analysis was performed on three heterogeneous GBM patient-derived low-passage cultures; this number is in line with previously published reports of similar datasets (e.g., 2-4). The derepression observed for putative targets is very mild, 1.2-2 fold (linear scale), which is also consistent with a fine-tuning mode of miRNA regulation (5,6). To validate the array results for genes of interest, we performed gene-specific qRT-PCR analysis on additional cell lines. These experiments demonstrated consistent derepression of 8 mRNA processing/splicing factors (MBNL1-MBNL3, SART3, RSRC1, SRSF11, PTBP2 and DGCR14) upon inhibition of miR-10b in all six glioma cell lines and GSC tested. The results of this validation are shown in the revised Figure 4B.

2. The genes are annotated by gene ontology and described as being enriched for cell cycle genes, which was known before and having functions in RNA processing and RNA splicing. Given the latter is not easy to functionally validate they check if a selected set of genes is in general more highly or more lowly expressed than in normal brain in publically available datasets (Fig3). This is not really pertinent! These analyses should be replaced by the correlation of the genes of interest with miR10b in the TCGA GBM and LGG data-sets where miR profiles are available. On this occasion the TCGA data could also be used to determine if the genes identified in the screen were the top correlated genes, or whether there are other gene sets that may be more relevant (statistically, biologically).

Response: Since the RNA processing/splicing, along with cell cycle, is the bioterm most significantly correlating with miR-10b in GBM TCGA (Table EV2), and RNA processing related genes are strongly enriched among those regulated by anti-miR-10b in GSCs (Table EV1 and Fig. 2B), we attempted to identify RNA processing related miR-10b direct targets.

In regard to the expression correlation of miR-10b/individual mRNA targets in the GBM TCGA, we now include this information as well (Appendix Table S1), but must note that it has to be interpreted with caution and used to neither assume nor rule out the direct regulation without additional experimental support. The tumor tissue analysis provides an end-stage snapshot of gene expression in a heterocellular specimen, and does not reflect the functional relationship. Our bioinformatics analysis of expression correlation between miR-10b and its previously experimentally validated targets (in LGG and GBM TCGA datasets) indicates a weak but significant inverse correlation with some targets (e.g. BCL2L11 and CYLD), but lack of such for the majority of them (CDKN1A, CDKN2A, HOXD10, PTCH1, Notch1, etc) (7-9).

Expression of three targeted splicing factors identified in this study, MBNL1, MBNL2, and DGCR14, showed significant inverse correlation with miR-10b; other factors revealed lack of such correlation (Appendix Table S1). In this context, we find Figure 3 informative and prefer to keep it included in the manuscript.

3. Figure S4 claims to identify alternatively spliced genes due to knock-down of miR10b, I don't think that the Affymetrix chip used is an appropriate tool to detect alternative splicing. At least it would call for respective validation.

Response: We agree with the reviewer that the utilized Affymetrix chips were not designed for identification of alternatively spliced mRNA isoforms. Nevertheless, these arrays contain multiple

probe-sets per gene, with numerous probe-sets detecting specific splice variants, and our analysis suggested significant effects on alternative splicing. To validate this observation, we confirmed differential effects of miR-10b inhibition on alternative splice isoforms of selected genes by qRT-PCR analysis. The qRT-PCR primers were designed that each pair of primers detects a specific splice variant of a gene. This analysis confirmed the microarray results and the new data is included in Appendix Figure S3 and page 5.

4. The genes followed up on that may be modulated by miR10b using reporter experiments would profit from validation in the TCGA data as suggested above.

Response: As described above, we included TCGA correlations between miR-10b and several regulated splicing factors in Appendix Table S1.

5. None of the genes is followed up on for biological evaluation and relevance for the observed phenotype.

Response: Based on the expression profiling, we believe that the observed phenotype of GSC treated anti-miR-10b is a result of coordinated mild regulation (fine-tuning) of numerous genes rather than strong regulation of a few targets. Similar mechanisms are commonly observed for other miRNAs in various studies (5,6). The reconstitution of such mechanism in vitro would require fine tuning levels of multiple genes, which might not be technically feasible. In this context, it would be difficult to rationalize the focus on a single/few individual target genes and, we believe, the results of such attempt would be rather scientifically misleading.

We should also note that, with multiple cell type/cell line specific miR-10b targets identified in this and prior studies (7-10), our primary goal was NOT to identify one or a few key targets, but rather discover common targets modulated in heterogeneous GBM cells and tissues that previous studies failed to identify and that would serve as potential biomarkers/readouts for the efficacy of therapeutic miR-10b inhibition. Indeed, the data shown in Figures 5-6 demonstrate strong response of MBNL1, MBNL2, and other identified factors to miR-10b inhibition in vivo. We now include this idea in the Discussion (page 14).

6. The in vivo experiments could profit from using several lines with distinct genotypes, or for the matter of testing migration using a truly invasive model. Figure 8 suggests that the model used is not really invasive.

Response: In this work, we utilized distinct human GBM8 xenograft and mouse GL261 allograft orthotopic GBM models, and three delivery modalities for the antagomir. The revised version of the manuscript includes a new systemic treatment experiment on GL261 models and provides important insights regarding the safety of the systemic treatment (Figure EV5, pages 10 and 15). We selected this model over human GSC-derived xenografts since it allows examination of systemic effects of the miR-10b inhibition in the immunocompetent environment. Of note, additional U87 s.c. glioma model was examined in our prior work (7). Since miR-10b inhibition impairs cell growth of all tested glioma cultures in vitro and experiments on intracranial GBM are both time-consuming and very expensive, and within the limited timeframe for the manuscript resubmission, we refrain from adding more test models to this manuscript.

Regarding the invasiveness of GBM8-derived tumors - those are among the most invasive xenograft models (11). The image shown in Fig. 8E does not represent this characteristic since the least invasive areas with relatively defined boundary have to be selected to quantify the number of invading cells.

Minor comments

There are references in the text to tables in the supplement with gene lists. The tables have no numbers or title, and it is not clear what they comprise. At present they are not very helpful.

We have removed original Supplementary files (that contained microarray data) from the revised manuscript. The microarray expression datasets associated with this manuscript have been deposited to GEO database and are publicly available.

REFERENCES:

1. Wakimoto H, Kesari S, Farrell CJ, Curry WT, Jr., Zaupa C, Aghi M, Kuroda T, Stemmer-Rachamimov A, Shah K, Liu TC, et al. Human glioblastoma-derived cancer stem cells: establishment of invasive glioma models and treatment with oncolytic herpes simplex virus vectors. *Cancer Res.* 2009;69(8):3472-81.
2. Chen J, Li JL, Chen Z, Griffin JD, Wu L. Gene expression profiling analysis of CRTCl-MAML2 fusion oncogene-induced transcriptional program in human mucoepidermoid carcinoma cells. *BMC Cancer.* 2015;15:803.
3. Eyles CE, Wu Q, Yan K, MacSwords JM, Chandler-Militello D, Misuraca KL, Lathia JD, Forrester MT, Lee J, Stamler JS, Goldman SA, Bredel M, McLendon RE, Sloan AE, Hjelmeland AB, Rich JN. Glioma stem cell proliferation and tumor growth are promoted by nitric oxide synthase-2. *Cell.* 2011; 146(1):53-66.
4. Varambally S, Dhanasekaran SM, Zhou M, Barrette TR, Kumar-Sinha C, Sanda MG, Ghosh D, Pienta KJ, Sewalt RG, Otte AP, Rubin MA, Chinnaiyan AM. The polycomb group protein EZH2 is involved in progression of prostate cancer. *Nature.* 2002; 419(6907):624-629.
5. Selbach M, Schwanhäusser B, Thierfelder N, Fang Z, Khanin R, Rajewsky N. Widespread changes in protein synthesis induced by microRNAs. *Nature.* 2008; 455(7209): 58-63.
6. Jost D, Nowojewski A, Levine E. Regulating the many to benefit the few: role of weak small RNA targets. *Biophys J.* 2013; 104(8):1773-1782.
7. Gabriely G, Yi M, Narayan RS, Niers JM, Wurdinger T, Imitola J, Ligon KL, Kesari S, Esau C, Stephens RM, Tannous BA, Krichevsky AM. Human glioma growth is controlled by microRNA-10b. *Cancer Res.* 2011; 71(10):3563-3572.
8. Sasayama T, Nishihara M, Kondoh T, Hosoda K, Kohmura E. MicroRNA-10b is overexpressed in malignant glioma and associated with tumor invasive factors, uPAR and RhoC. *Int J Cancer.* 2009;125(6):1407-1413.
9. Lin J, Teo S, Lam DH, Jeyaseelan K, Wang S. MicroRNA-10b pleiotropically regulates invasion, angiogenicity and apoptosis of tumor cells resembling mesenchymal subtype of glioblastoma multiforme. *Cell Death Dis.* 2012; 3:e398.
10. Teplyuk NM, Uhlmann EJ, Wong AH, Karmali P, Basu M, Gabriely G, Jain A, Wang Y, Chiocca EA, Stephens R, Marcusson E, Yi M, Krichevsky AM. MicroRNA-10b inhibition reduces E2F1-mediated transcription and miR-15/16 activity in glioblastoma. *Oncotarget.* 2015; 6(6):3770-3783.
11. Wakimoto H, Mohapatra G, Kanai R, Curry WT Jr, Yip S, Nitta M, Patel AP, Barnard ZR, Stemmer-Rachamimov AO, Louis DN, Martuza RL, Rabkin SD. Maintenance of primary tumor phenotype and genotype in glioblastoma stem cells. *Neuro Oncol.* 2012; 14(2):132-144.

2nd Editorial Decision

21 December 2015

Thank you for the submission of your revised manuscript to EMBO Molecular Medicine. We have now received the enclosed report from the referee asked to re-assess it. As you will see this reviewer is now globally supportive and I am pleased to inform you that we will be able to accept your manuscript pending final editorial amendments.

I look forward to reading a new revised version of your manuscript as soon as possible.

***** Reviewer's comments *****

Referee #1 (Remarks):

The authors have successfully addressed all concerns that I had with the previous submission. I have no further request. The paper has been significantly improved.